# A DNA replication-independent function of pre-replication complex genes during cell invasion in *C. elegans*

**Evelyn Lattmann**[1☉], **Ting Deng**[1,2☉], **Michael Walser**[1☉], **Patrizia Widmer**[1],
**Charlotte Rexha-Lambert**[1], **Vibhu Prasad**[1¤a], **Ossia Eichhoff**[3], **Michael Daube**[1],
**Reinhard Dummer**[3], **Mitchell P. Levesque**[3], **Alex Hajnal**[1]*

**1** Department of Molecular Life Sciences, University of Zürich, Zürich, Switzerland, **2** Molecular Life Science PhD Program, University and ETH Zürich, Zürich, Switzerland, **3** Department of Dermatology, University Hospital Zürich, Zürich, Switzerland

☉ These authors contributed equally to this work.
¤a Current address: Department of Infectious Diseases, Molecular Virology, Heidelberg University, Heidelberg, Germany
* alex.hajnal@mls.uzh.ch

**Data Availability Statement:** Numerical values to generate the graphs can be found in the S1 Data and S2 Data files. The RNAseq data are accessible at NCBI Gene Expression Omnibus (GEO) (http://

## Abstract

Cell invasion is an initiating event during tumor cell metastasis and an essential process during development. A screen of *C. elegans* orthologs of genes overexpressed in invasive human melanoma cells has identified several components of the conserved DNA pre-replication complex (pre-RC) as positive regulators of anchor cell (AC) invasion. The pre-RC genes function cell-autonomously in the G1-arrested AC to promote invasion, independently of their role in licensing DNA replication origins in proliferating cells. While the helicase activity of the pre-RC is necessary for AC invasion, the downstream acting DNA replication initiation factors are not required. The pre-RC promotes the invasive fate by regulating the expression of extracellular matrix genes and components of the PI3K signaling pathway. Increasing PI3K pathway activity partially suppressed the AC invasion defects caused by pre-RC depletion, suggesting that the PI3K pathway is one critical pre-RC target. We propose that the pre-RC, or a part of it, acts in the postmitotic AC as a transcriptional regulator that facilitates the switch to an invasive phenotype.

## Introduction

Cell invasion, defined as the movement of cells across compartment boundaries formed by basement membranes (BMs), is an essential process during normal development. Yet, cell invasion can have fatal consequences during cancer progression [1]. During metastasis, individual cells of a primary tumor adopt an invasive phenotype, resulting in metastatic tumor spreading through the vasculature. Despite the different outcomes, developmental and cancerous cell invasion have many common features. Both types of invasion involve a phenotypic switch called epithelial to mesenchymal transition (EMT). Invasive cells adopt a mesenchymal-like phenotype characterized by the formation of actin-rich protrusions called invadopodia, the loss of cadherin-mediated cell adhesion and the expression of metalloproteinases

www.ncbi.nlm.nih.gov/geo/) under accession number GSE149523, and the flow cytometry data can be found in the FlowRepository (https://flowrepository.org/) under accession number FR-FCM-Z4XE.

**Funding:** This work was supported by grants from the Swiss National Science Foundation no. 31003A-166580 to A.H., a grant from the Swiss Cancer league no. 4377-02-2018 to A.H., and the University Research Priority Project in Translational Cancer Research. The funders had no role in study design, data collection and analysis, decision to publish, or preparation of the manuscript.

**Competing interests:** The authors have declared that no competing interests exist.

**Abbreviations:** AC, anchor cell; ACEL, AC-specific *lin-3* enhancer element; BM, basement membrane; Dox, doxycycline; EMT, epithelial to mesenchymal transition; FGCZ, Functional Genomics Center Zürich; FLP, flippase; FRT, FLP recognition target sequence; GEO, Gene Expression Omnibus; HRP, horseradish peroxidase; HU, hydroxyurea; MMP, metalloproteinase; pre-IC, pre-initiation complex; pre-RC, pre-replication complex; Pvl, protruding vulva; Thy, thymidine; VPC, vulval precursor cell; VU, ventral uterine.

(MMPs) that break down the BM. This phenotypic switch can be stimulated by extracellular signals, such as TGF-β, EGF, and Wnt, and is regulated by conserved zinc finger transcription factors of the Twist, Snail, and Krüppel family as well as by the Jun/Fos (AP-1) transcription factors that activate characteristic, proinvasive transcriptional programs [2].

Here, we have used *C. elegans* vulval development as an in vivo model to investigate the regulation of cell invasion. During the development of the vulva, the egg-laying organ of the hermaphrodite, the anchor cell (AC) in the ventral uterus breaches 2 BMs and invades the underlying vulval epithelium to establish direct contact between the developing uterus and vulva [3,4]. In the wild type, AC invasion occurs during a specific time period, beginning in mid-L3 larvae after the first round of vulval cell divisions (the Pn.p 2-cell stage) and ending after the second round of divisions has been completed (Pn.p 4-cell stage). The Netrin homolog UNC-6 secreted by neurons in the ventral nerve cord, together with an unknown cue released by the primary (1˚) vulval precursor cells (VPCs) that are nearest to the AC, polarizes and guides the invading AC ventrally [5]. Activation of the UNC-40/DCC receptor in the AC induces the recruitment of several actin regulators and the αβ-integrin complex to the ventral AC cortex, thereby creating an invasive membrane [6]. At the same time, the expression of proinvasive genes is activated by the AP-1 transcription factor FOS-1A and *egl-43*, which encodes a zinc finger transcription factor homologous to the mammalian EVI1 proto-oncogene [7–9]. Known target genes of FOS-1A and EGL-43/EVI1 in the AC include MMP *zmp-1*, protocadherin *cdh-3*, and hemicentin *him-4* [7].

Another important aspect of cell invasion is the dichotomy between the proliferative and invasive state of a cell. The AC must remain arrested in the G1 phase of the cell cycle in order to activate proinvasive gene expression and breach the BMs [8]. The histone deacetylase *hda-1*, the homolog of mammalian HDAC1, the nuclear hormone receptor *nhr-67* and *egl-43* are critical to establish the G1 arrest of the AC [9,10]. Likewise, the invasiveness of human cancer cells often correlates with proliferation arrest. For example, during metastatic progression, melanoma cells can undergo multiple rounds of phenotype switching between a proliferative and an invasive state [11]. However, the mechanisms by which cell cycle regulators affect cell invasion are poorly understood.

The conserved pre-replication complex (pre-RC) plays an essential role during the G1 phase of the cell cycle to license the DNA replication origins that will be used in the following S-phase [12–14]. The pre-RC assembles during the late M and early G1 phase of the cell cycle by the sequential recruitment of the origin recognition complex proteins ORC1–ORC6 to DNA replication origins, followed by the CDC6 protein together with the chromatin licensing and replication factor CDT1, and finally, the DNA helicase complex formed by the mini-chromosome maintenance proteins MCM2 to MCM7 [12,15]. After the pre-RC has been assembled in the G1 phase, the origins are licensed to initiate DNA replication during the following S-phase. Thereafter, CDK/Cyclin activity in the late G1 and early S-phase induces the disassembly and partial degradation of the pre-RC to prevent the relicensing of replication origins.

Using *C. elegans* AC invasion as a model to screen for genes controlling cell invasion, we have found that multiple components of the pre-RC are required for BM breaching by the AC. In addition, we have identified the PI3K pathway as one critical downstream target of the pre-RC promoting BM breaching by the AC. Together, these data point to a replication-independent function of the pre-RC in regulating cell invasion during the G1 phase of the cell cycle.

# Results

## Identification of cell invasion regulators among differentially expressed genes in invasive melanoma cells

In order to identify genes that are functionally involved in regulating cell invasion, we investigated candidate genes whose orthologs were up-regulated in human melanoma cells that had

been stimulated with TGF-β, exposed to hypoxia, or which overexpressed the transcription factor SOX9, and included genes up-regulated in migrating neural crest cells that had undergone EMT [16–18]. This resulted in a list of 104 differentially expressed human genes that were up-regulated under at least 2 conditions and for which a *C. elegans* ortholog could be identified (**S1 Table**).

RNAi-mediated knock-down of 32 of the 104 *C. elegans* orthologs resulted in a protruding vulva (Pvl) phenotype, which is characteristic of defects in vulval morphogenesis or in AC invasion. In order to specifically identify regulators of AC invasion, we scored BM breaching by the AC after RNAi-mediated knock-down. AC invasion was analyzed in L3 larvae after the 1˚ VPCs had divided twice, at the VPC 4-cell (Pn.pxx) stage of vulval development. If viability and fertility permitted, we scored the F1 progeny of RNAi-treated mothers, or else we scored the P0 animals that had been exposed to the dsRNA-producing bacteria from the L1 stage on. To visualize BM breaching by the invading AC, we used a LAM-1::GFP reporter that labels the 2 BMs separating the AC from the underlying vulval cells (**Fig 1A**) [5]. Among the 104 candidates tested, RNAi knock-down of 18 genes perturbed AC invasion (**S1 Table**). Notably, 3 genes required for normal AC invasion encode cell cycle regulators; *cdc-6* encodes an essential component of the DNA pre-RC [12,15], *cyd-1* encodes the only Cyclin D homolog in *C. elegans* [19], and *cdk-12* encodes the Cyclin-dependent kinase required for activation of RNA polymerase II [20]. In particular, Cyclin D was shown to be a driver for invasion in a study investigating the development of thin melanoma [21]. After RNAi knock-down of *cdc-6*, the AC failed to invade in 32% of the cases (**Fig 1A**).

In the following, we focused our experiments on studying the role of the pre-RC genes during cell invasion.

## Several components of the pre-replication complex regulate AC invasion

Since the AC remains arrested in the G1 phase of the cell cycle while it breaches the BMs [8], the identification of the pre-RC component CDC-6 as a regulator of AC invasion was unexpected. We therefore tested if other pre-RC components are also required for BM breaching by the AC. Besides *cdc-6*, RNAi-mediated knock-down of several *orc* genes, as well as of *cdt-1* and *mcm-7* led to BM breaching defects (**Fig 1A** and **Table 1**). The penetrance of the invasion defects ranged from 15% for *cdt-1* to 51% upon *orc-2* RNAi (not taking into account the single cases observed after *orc-3* and *orc-4* RNAi). Among the *mcm* genes, only *mcm-7* RNAi resulted in an invasion defect in 34% of the animals (**Fig 1A** and **Table 1**). However, the absence of AC invasion defects for other *mcm* genes tested could be due to low efficiencies of the RNAi knock-down.

Thus, several components of the pre-RC promote BM breaching by the AC. In addition to the AC invasion defects, RNAi of pre-RC complex components induced a proliferation arrest in different cells of the larvae, notably in the dividing VPCs adjacent to the AC (**Fig 1B**).

## The DNA replication pre-initiation complex is not required for AC invasion

At the onset of the S-phase, the pre-RC transitions into the pre-initiation complex (pre-IC) through the recruitment of accessory proteins of the GINS complex (PSF-1, PSF-2, PSF-3, SLD-5) and additional factors, such as CDC-45, MCM-10, CDC-7, and PCN-1, which are necessary to initiate DNA replication [12,15,22]. In order to test if the initiation of DNA replication is required for AC invasion, we examined the AC invasion phenotype of a *psf-1* null *(lf)* allele and performed RNAi against different pre-IC components. Except for a single case in *cdc-45* RNAi-treated animals, we did not observe any invasion defects after RNAi knock-down or in mutants of pre-IC genes (**Fig 1A** bottom row and **S2 Table**). In some animals, the gap the AC created in the BMs of *psf-1(lf)* mutants was not expanded as the underlying VPCs had

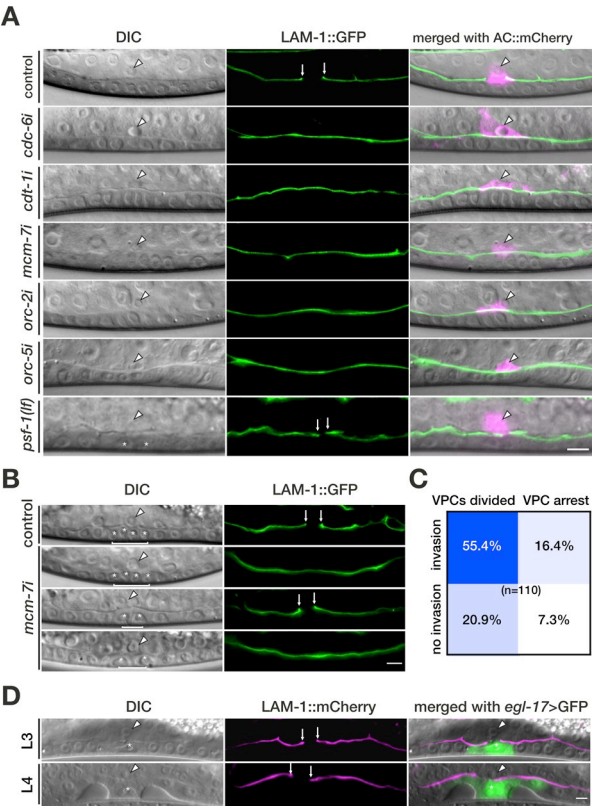

**Fig 1. Several pre-RC components are required for AC invasion.** (A) DIC images (left panels), corresponding green fluorescence image of the LAM-1::GFP reporter (middle panels), and merged images including the AC marker mCherry::moeABD (*qyIs50*) in magenta (right panels). The arrowheads in the DIC and merged panels point at the AC nucleus, and the arrows indicate the sites of BM breaching. In control animals fed with the empty RNAi vector (top panels), the BM was breached at the Pn.pxx stage. RNAi knock-down of *cdc-6*, *cdt-1*, *mcm-7*, *orc-2*, and *orc-5* prevented BM breaching (see Table 1 for the numbers of animals scored). A *lf* mutation in the pre-IC component *psf-1* did not prevent BM breaching despite the VPC proliferation arrest (bottom panels; see S2 Table for the number of animals scored). (B) AC invasion and VPC proliferation defects occur independently. The 4 different combinations of phenotypes observed after empty vector control and *mcm-7* RNAi are shown; (from top to bottom) normal BM breaching and VPC proliferation, no BM breaching but VPC proliferation, BM breaching but VPC proliferation arrest, no BM breaching and VPC proliferation arrest. (C) Correlation between the AC invasion and VPC proliferation defects after *mcm-7* RNAi. The frequencies of the 4 combinations of phenotypes among 110 animals scored are shown inside the boxes. (D) Expression of CKI-1 in P6.p using the *zhEx334[egl-17>cki-1]* transgene blocked the division of P6.p as reported [24] (left panels) but did not prevent BM breaching (middle panels). BM breaching was visualized with the LAM-1::mCherry marker (*qyIs127*) and the 1° VPCs were labeled with the *ayIs4[egl-17>gfp]* reporter. An L3 larva at the Pn.pxx stage and an L4 larva during vulval morphogenesis are shown. The asterisks in the bottom panel of (A) and in (B) and (D) label the nuclei of the undivided P6.p(x) cells. The right panels show the merged DIC and LAM-1::mCherry images including the e*gl-17>*GFP signal in green. The scale bars are 5 μm. AC, anchor cell; BM, basement membrane; DIC, differential interference contrast; pre-IC, pre-initiation complex; pre-RC, pre-replication complex; VPC, vulval precursor cell.

ceased to proliferate (**Fig 1A** bottom row), though the average size of the BM gap was not significantly changed in *psf-1(lf)* mutants (6.8 ± 1.8 μm in the wild type versus 7.7 ± 4.5 μm in *psf-1(lf)*) [23].

We conclude that components of the pre-IC are not necessary for AC invasion.

## VPC proliferation is not required for AC invasion

Invasion by the postmitotic AC is linked to the differentiation and proliferation of the underlying 1° VPCs that produce guidance signals, which attract the AC ventrally [7]. It has previously

**Table 1. Knock-down of several pre-RC components prevents BM breaching.** The genes encoding different pre-RC components were analyzed by RNAi for their potential to breach the BM at the P6.p.xx stage as described in **Fig 2**. The observed BM breaching defects are indicated in absolute numbers. Pvl indicates a protruding vulva and Ste a sterile phenotype at the adult stage.

| Sequence name | Gene name | Phenotype | Invasion defects/animals scored |
|---|---|---|---|
| Origin of replication components | | | |
| Y39A1A.12 | *orc-1* | Pvl, Ste | 0/20 |
| F59E10.1 | *orc-2* | Pvl, Ste | 26/51 |
| Y119D3B.11 | *orc-3* | - | 1/20 |
| Y39A1A.13 | *orc-4* | - | 1/20 |
| ZC168.3 | *orc-5* | Pvl, Ste | 6/29 |
| Mini chromosome maintenance complex components | | | |
| Y17G7B.5 | *mcm-2* | Ste | 0/20 |
| C25D7.6 | *mcm-3* | Ste | 0/20 |
| Y39G10AR.14 | *mcm-4* | Pvl, Ste | 0/20 |
| R10E4.4 | *mcm-5* | Ste | 0/20 |
| ZK632.1 | *mcm-6* | Ste | 0/20 |
| F32D1.10 | *mcm-7* | Pvl, Ste | 43/126 |
| Replication-licensing factors | | | |
| Y54E10A.15 | *cdt-1* | Pvl, Ste | 10/65 |
| C43E11.10 | *cdc-6* | Pvl, Ste | 13/56 |

BM, basement membrane; pre-RC, pre-replication complex.

been shown that arresting VPCs in S-phase by hydroxyurea (HU) treatment does not prevent AC invasion [7]. However, in that experiment, the timing of invasion could not be assessed relative to individual VPC fates, since HU treatment arrested the division of all VPCs. We therefore tested whether the AC invasion defects observed after down-regulation of pre-RC components might be caused by a proliferation arrest of the VPCs, and if preventing the 1° VPCs from entering into S-phase could inhibit AC invasion. We first examined on an animal per animal basis whether the AC invasion defects caused by *mcm-7* RNAi correlated with the proliferation arrest of the VPCs. An analysis of 110 animals revealed no correlation, as *mcm-7i* invasion defects occurred independently of whether the VPCs had divided or not (**Fig 1B and 1C**). Next, we blocked the cell cycle of the 1° VPC (P6.p) in the G1 phase by overexpressing the CDK inhibitor *cki-1* under control of the 1° lineage-specific *egl-17* promoter (*zhEx334[egl-17>cki-1::gfp; myo-2::mCherry]* [24] and used a LAM-1::mCherry reporter to score BM breaching. In all *egl-17>cki-1::gfp; lam-1>lam-1::mCherry* animals with an undivided P6.p cell at the P5/7.p 4-cell ($n = 16$) or 8-cell stage ($n = 16$) the BMs were breached, indicating that AC invasion does not depend on S-phase entry of the 1° VPC (**Fig 1D**).

We conclude that entry of the 1° VPCs into S-phase is not required for AC invasion and, therefore, the AC invasion defects caused by *mcm-7* RNAi are not due to a VPC cell-cycle arrest.

## *mcm-7* is expressed in the AC prior to invasion

In order to determine the expression pattern of the pre-RC components during AC invasion, we focused on *mcm-7*, because it encodes a core subunit of the MCM complex that is the last component recruited to the pre-RC. We used CRISPR/Cas9 engineering to insert a *gfp* tag upstream of the translational start codon in the *mcm-7* locus along with 2 flippase (FLP) recognition target (FRT) sites within the *gfp* cassette. This permitted the tissue-specific inactivation of *mcm-7* by FLP-induced recombination (*zh118[frt::gfp::mcm-7]*, **S1 Fig**) [25,26]. *zh118[frt::*

*gfp*::*mcm-7]* animals exhibited GFP::MCM-7 expression in the nuclei of proliferating cells, including many uterine cells and the VPCs (**Fig 2A**). In mid-L2 larvae, GFP::MCM-7 was expressed at comparably lower levels in the newly specified AC. During the late L2 and early L3 stage, GFP::MCM-7 expression in the AC further declined but was still detectable in mid-L3 larvae just prior to invasion. In late L3 larvae, after the BM had been breached, no GFP::MCM-7 expression was detectable in the AC (**Fig 2A**). Besides MCM-7, also CDT-1::GFP expression was consistently observed in the AC nucleus prior to and during invasion (**S2 Fig**). In contrast to GFP::MCM-7, CTD-1::GFP continued to be expressed in the AC after BM breaching in late L3 larvae (Pn.pxx stage). The adjacent uterine cells showed predominantly cytoplasmic CDT-1::GFP expression.

Since GFP::MCM-7 expression levels were highest in the newly formed AC, we tested whether *mcm-7* is required for the specification or function of the AC during vulval induction. We first examined the expression of the b-HLH transcription factor HLH-2, whose expression onset in early L2 larvae instructs the AC fate [27]. *mcm-7* RNAi had no significant effect on HLH-2::GFP expression levels in the AC of mid-L3 larvae (**S3A and S3B Fig**). The differentiated AC then expresses the proto-cadherin *cdh-3* [28] and the EGF homolog *lin-3*, which induces the differentiation of the adjacent VPCs [4,29]. A transcriptional *cdh-3>gfp* reporter was not changed after *mcm-7* RNAi, while *lin-3>gfp* expression was reduced (**S3C–S3F Fig**). However, the *lin-3* levels in the AC after *mcm-7* RNAi were sufficient to induce vulval differentiation, as the 1° vulval cell fate marker *egl-17>gfp* [30] was expressed in the 1° VPCs of all *mcm-7* RNAi animals (**S3G Fig**). Thus, MCM-7 is not necessary for the specification of the AC fate or for the normal function of the AC during the induction of vulval differentiation.

## *mcm-7* acts in the AC to promote BM breaching

The observation that a proliferation arrest of the VPCs did not impair BM breaching raised the possibility that *mcm-7* acts cell-autonomously in the AC to promote invasion, besides its established role in replication origin licensing in proliferating cells. To test whether the low GFP::MCM-7 expression levels observed in the AC of L3 larvae promote invasion, we first performed Pn.p cell and AC-specific RNAi of *mcm-7* [31–33]. Notably, 16% of the animals treated with AC-specific *mcm-7* RNAi showed a BM breaching defect, whereas none of the control RNAi-treated or Pn.p cell-specific *mcm-7* RNAi animals exhibited BM breaching defects (**Fig 2B**). This analysis suggested a cell-autonomous role of *mcm-7* in the postmitotic AC. To independently confirm these findings, we performed FLP/FRT-induced mosaic analysis in the *zh118[frt::gfp::mcm-7]* strain [26]. The FLPase was expressed under control of the heat shock–inducible *hs-16-48* promoter (*hs>flp*) and induced by a single heat shock. Two cells with the potential to become the AC (Z1.ppp and Z4.aaa) are born during the third round of gonadal cell division in the mid to late L1 stage [27]. During the L2 stage, one of these 2 precursor cells adopts the AC fate, while the other cell becomes the ventral uterine (VU) precursor cell. To test if MCM-7 functions in the AC or in its mother cell (Z1.pp or Z4.aa), freshly hatched L1 worms were grown until the late L1/early L2 stage before heat shock induction of the FLPase (see Materials and methods). This protocol resulted in the development of mosaic animals that had lost GFP::MCM-7 expression either in most uterine cells, in the VPCs, or in both lineages (**Fig 2C**). Among the VPC mosaics, we focused on animals showing loss of GFP::MCM-7 expression in the 1° lineage because the 1° VPCs are necessary to induce AC invasion [7]. Around 29% of animals with a loss of GFP::MCM-7 expression in most uterine cells and 38% of animals lacking expression in both the uterus and the 1° VPCs showed a BM breaching defect (**Fig 2C** and white bars in **2D**). Except for the AC, other uterine cells are not required for BM breaching [34]. By contrast, only 3.1% of the 1° VPC mosaics and 2.7% of the cases

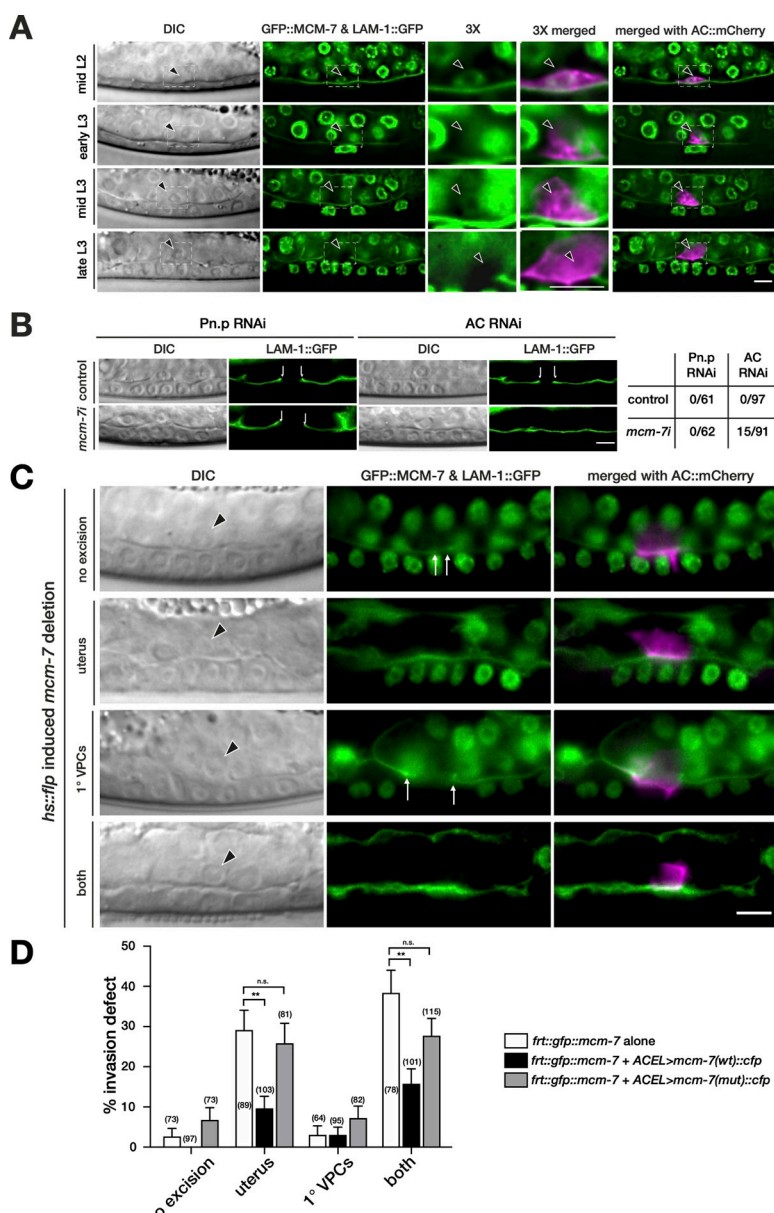

**Fig 2. *mcm-7* acts in the AC to induce BM breaching.** (A) GFP::MCM-7 (*zh118*) and LAM-1::GFP expression beginning at the mid-L2 stage, shortly after AC specification, until the late-L3 (P6.pxx) stage after BM breaching are shown. DIC images are shown in the left panels; GFP::MCM-7 and LAM-1::GFP expression in green are shown in the middle panels. The dashed boxes indicate the areas around the AC shown at 3-fold magnification in the center panels. The right panels show the GFP images merged with the AC marker *ACEL>mCherry* (*zhIs127*) in magenta. The arrowheads point at the AC nuclei. (B) Pn.p and AC-specific RNAi of *mcm-7* was performed as described in Materials and methods. The top row shows DIC and LAM-1::GFP expression in empty vector–fed control animals and the bottom row *mcm-7* RNAi-treated animals at the Pn.p xx stage. BM breaching is indicated by the arrows. The table to the right shows the occurrence of BM breaching defects in absolute numbers. (C) FLP/FRT-mediated *mcm-7* mosaic analysis using the *zh118(frt:gfp::mcm-7)* allele with heat shock induced *flp* expression to excise *mcm-7*, as described in Materials and methods and S1 Fig. DIC images are shown in the left panels, LAM-1::GFP expression used to score BM breaching together with GFP::MCM-7 in the middle panels, and the merged images including the AC marker *cdh-3> mCherry::PLCδPH* (*qyIs23*) (magenta) in the right panels. The top row shows a control animal at the Pn.pxx stage that did not show any GFP::MCM-7 excision. Examples for the 3 major classes of mosaic animals observed are shown underneath; excision in most uterine cells, excision in the 1° VPCs, and excision in both tissues. The arrows indicate the sites of BM breaching and the arrowheads point at the AC nuclei. (D) Quantification of the AC invasion defects in the different classes of FLP-induced *zh118(frt::gfp::mcm-7)* mosaic animals carrying no transgene, the wild-type, or the mutant rescue transgenes. The 3 major classes of mosaic animals and control animals without *mcm-7* excision were

scored separately for each genotype. The data shown are the combined results obtained in 3 biological replicates. n refers to the number of mosaic animals scored, and the error bars indicate the SEM. *p*-Values were determined with a nonparametric Kruskal–Wallis test with Dunn's multiple comparison correction and are indicated as ** for $p < 0.01$ and n.s. for $p > 0.5$. See S1 Data for the numerical values used to generate the graph. The scale bars are 5 μm. AC, anchor cell; BM, basement membrane; DIC, differential interference contrast; FLP, flippase; FRT, FLP recognition target sequence; VPC, vulval precursor cell.

without an apparent loss of GFP::MCM-7 expression in any of the 2 tissues exhibited a BM breaching defect.

Taken together, the tissue-specific RNAi experiments and the mosaic analysis indicated that *mcm-7* acts cell-autonomously in the postmitotic AC to positively regulate BM breaching. Though, the mosaic analysis could not distinguish if *mcm-7* functions while the AC is adopting its fate or afterwards. It should also be noted that the highest penetrance of BM breaching defect after FLP/FRT-mediated inactivation of *mcm-7* was 38%, suggesting that pre-RC activity may not be absolutely required for AC invasion.

## The DNA helicase activity of the pre-RC is required for AC invasion

The MCM protein complex provides the helicase activity of the pre-RC that is necessary for DNA unwinding during the formation of a replication fork. To test if the MCM helicase activity is necessary for AC invasion, we introduced 4 point mutations ($K_{396}S_{397} \rightarrow AA$ and $D_{454}E_{455} \rightarrow AA$) in the strongly conserved ATPase motif of MCM-7 (**S4A Fig**). These mutations have previously been shown to eliminate the helicase activity of the human MCM-4/6/7 complex [35]. By expressing the wild-type or mutant MCM-7 proteins tagged with CFP under control of the AC-specific *lin-3* enhancer element (ACEL) [36] and the *pes-10* minimal promoter (*zhIs119[ACEL>mcm-7(wt)::cfp]* and *zhIs142[ACEL>mcm-7(mut)::cfp]*) and simultaneously eliminating endogenous MCM-7 activity using the *zh118[frt::gfp::mcm-7]* allele to create *hs>flp*-induced mosaics, we could selectively assay MCM-7 function in the AC. MCM-7::CFP expression was first observed in late L1/early L2 larvae in the differentiating AC, and expression became more robust during the mid to late L2 stage after the AC had adopted its fate (**S4B and S4C Fig**). Remarkably, both wild-type and mutant rescue transgenes that were integrated as single copies at the same chromosomal site by MosSci [37] showed relatively faint MCM-7::CFP expression despite the use of the strong *lin-3 ACEL* enhancer element (**S4B Fig**). Therefore, MCM-7 expression appears to be down-regulated in the AC at the posttranscriptional level. Nevertheless, the wild-type *ACEL>mcm-7(wt)::cfp* transgene partially rescued the AC invasion defect of *mcm-7* mosaic animals (**Fig 2D**, black bars). Since no ACEL>MCM-7::CFP expression was observed before the birth of the AC during the L1 stage, these data further support the conclusion of the mosaic analysis and tissue-specific RNAi experiments, which indicated that MCM-7 acts cell autonomously in the postmitotic AC. By contrast, the helicase-deficient transgene *ACEL>mcm-7(mut)::cfp* failed to restore the BM breaching activity to the AC (**Fig 2D**, gray bars).

Therefore, the MCM helicase activity is not only required for the initiation of DNA replication, but also for BM breaching by the AC. Alternatively, the ATP-binding motif in MCM-7 may perform another MCM helicase-independent function during AC invasion.

## *mcm-7* controls BM breaching in the G1-arrested AC

It was previously reported that the AC must remain arrested in the G1 phase of the cell cycle in order to adopt the invasive cell fate [8]. We therefore examined whether the pre-RC controls the cell cycle state of the AC by examining the expression of cell cycle reporters. A

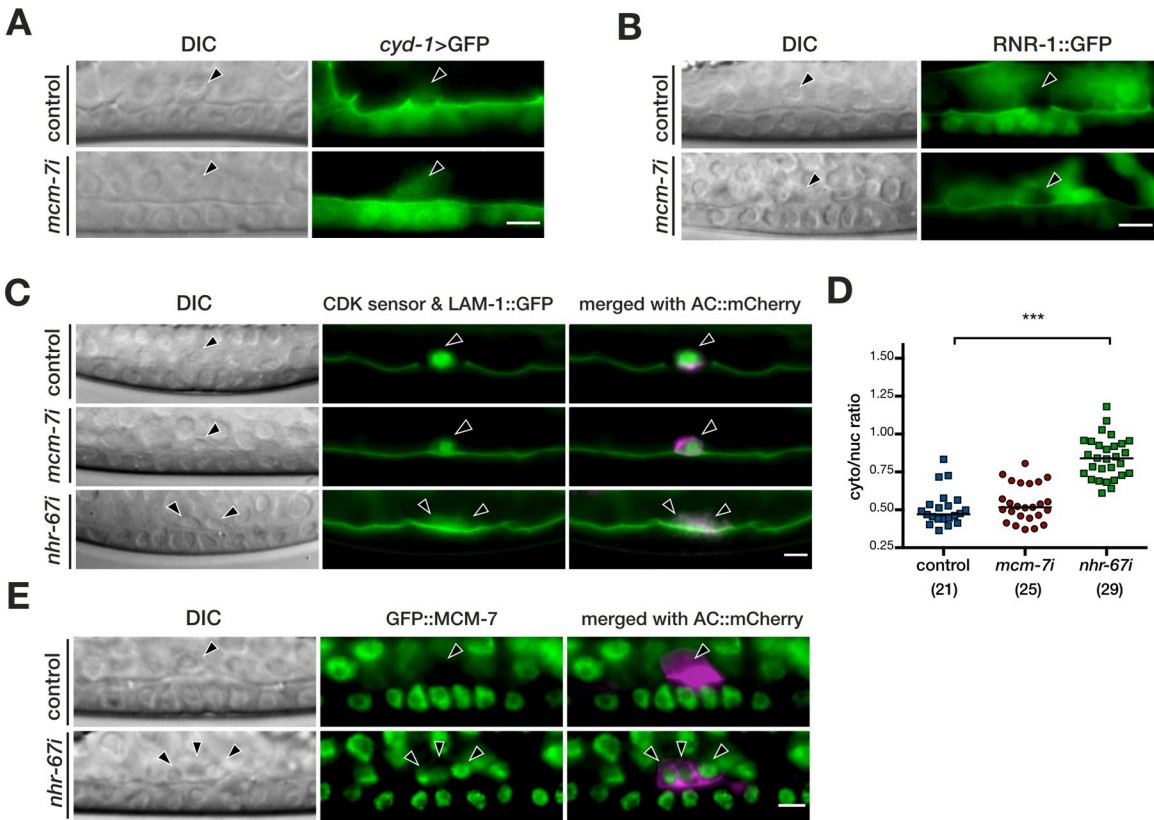

**Fig 3. *mcm-7* controls invasion in the G1-arrested AC.** (A) Expression of the G1-phase marker *cyd-1>gfp* in an empty vector–treated control (top row, *n* = 19) and after *mcm-7* RNAi (bottom row, *n* = 25). (B) Expression of the S-phase marker RNR-1::GFP in an empty vector–treated control (top row, *n* = 30) and after *mcm-7* RNAi (bottom row, *n* = 22). (C) AC-specific expression of the CDK activity sensor in an empty vector–treated control (top row), after *mcm-7* RNAi (middle row) and after *nhr-67* RNAi (bottom row). (D) Quantification of the CDK sensor activity as the cytoplasmic to nuclear GFP intensity ratios, as described in Materials and **m**ethods. The numbers of animals scored are indicated in brackets, and the horizontal lines indicate the median values. Statistical significance was determined with a two-tailed *t* test for independent samples and is indicated as *** for *p* < 0.001. See S1 Data for the numerical values used to generate the graph. (E) Strong GFP::MCM-7 in the **3** noninvasive ACs formed after *nhr-67* RNAi (bottom row) compared to the weak expression in the single AC of an empty vector–treated control animal (top row). The DIC images are shown in the left panels, the reporter expression in the middle panels, and in (C) and (E) the merged images together with the AC markers *ACEL>mCherry* (*zhIs127*) and *cdh-3>mCherry::PLCδPH* (*qyIs23*), respectively, in magenta are shown in the right panels. The arrowheads point at the AC nuclei. The scale bars are 5 μm. AC, anchor cell; DIC, differential interference contrast.

transcriptional reporter for the G1 phase marker *cyd-1 cyclinD* (*zhIs131[cyd-1>gfp])* was expressed in the AC of *mcm-7* RNAi-treated animals at similar levels as in control animals (**Fig 3A**). Conversely, the expression of the S phase marker *rnr-1::gfp* was undetectable in the AC of control and of *mcm-7* RNAi-treated animals (**Fig 3B**) [38]. Consistent with these findings, the histone deacetylase *hda-1* that is induced in the AC upon G1 arrest and is required for the invasive fate remained expressed after *mcm-7* knock-down (**S5 Fig**) [8].

In addition to the analysis of cell cycle markers, we made use of a CDK activity sensor to directly quantify CDK activity in the AC [39]. This sensor consists of a portion of the DNA Helicase B containing 4 CDK phosphorylation sites, a nuclear export, and a nuclear import signal, N-terminally fused to an eGFP tag. Upon phosphorylation by CDKs, the sensor translocates from the nucleus to the cytoplasm. Thus, a low cytoplasmic to nuclear ratio of the eGFP signal indicates weak CDK activity. For our purpose, we expressed the CDK sensor specifically in the AC under control of the *lin-3* ACEL enhancer/*pes-10* minimal promoter fragment (*zhIs130[ACEL>cdk sensor::egfp]*). In control animals, the average cytoplasmic to nuclear

fluorescence ratio in the AC was 0.51 ± 0.12 (**Fig 3C and 3D**). *mcm-7* RNAi did not cause a change in sensor activity (average fluorescence ratio 0.54 ± 0.12), indicating that the pre-RC is not required for the G1 arrest of the AC. By contrast, RNAi-mediated knock-down of the nuclear hormone receptor *nhr-67*, which is required for the G1 phase arrest of the AC [8], resulted in a significant increase in CDK activity (average ratio 0.84 ± 0.14) (**Fig 3C and 3D**). Moreover, reducing *nhr-67* levels resulted in the formation of multiple ACs that failed to invade, as reported previously [8], and expressed high levels of GFP::MCM-7 characteristic of proliferating cells (**Fig 3E**).

Hence, the down-regulation of *mcm-7* did not affect the G1 arrest of the AC. Low expression levels of MCM-7 are associated with G1 arrest and AC invasion, whereas high MCM-7 expression levels accompany the loss of the invasive phenotype in proliferating ACs.

## *mcm-7* regulates invadopodia formation and extracellular matrix gene expression in the AC

Prior to BM breaching, the AC is polarized ventrally and extends actin-rich protrusions (invadopodia) toward the 1˚ VPCs, which are aligned at the ventral midline [5]. To assess whether the pre-RC controls invadopodia formation, we quantified the polarity of an F-actin reporter after RNAi knock-down of *mcm-7*. The probe consisted of the F-actin-binding domain of moesin fused to *mCherry* and was expressed under control of the AC-specific *mk62-63* enhancer element of *cdh-3* (*mCherry::moeABD*) [6]. In control animals, the mCherry:: moeABD signal was polarized toward the ventral side and enriched at the tip of the invasive protrusion, where the AC breaches the BM (**Fig 4A and 4B**). Upon *mcm-7* RNAi treatment, the F-actin reporter was depolarized and no properly organized invasive membrane domain was formed by the AC (**Fig 4A and 4B**).

The invasive AC phenotype is characterized by the expression of several proinvasive genes, such as the matrix MMP *zmp-1* and the hemicentin *him-4*, which is secreted by the AC to remodel the BM [7]. RNAi knock-down of *mcm-7* decreased expression of the *him-4>ΔSP::gfp* and *zmp-1>cfp* reporters in the AC (**Fig 4C–4F**). The reduction in proinvasive gene expression was not limited to the pre-RC component *mcm-7* but was also observed after RNAi knock-down of *cdc-6* (**Fig 4G and 4H**). Since *zmp-1* and *him-4* expression are regulated by the AP-1 transcription factor *fos-1* and the zinc finger transcription factor *egl-43* [28,40], we analyzed the expression patterns of these 2 transcription factors. Neither *fos-1* nor *egl-43* expression was changed after *mcm-7* RNAi (**Fig 4I–4L**). These data suggested that the pre-RC acts downstream of or in parallel with *fos-1* and *egl-43* to activate *zmp-1* and *him-4* expression in the AC. In order to test whether *mcm-7* might be regulated by *fos-1*, we examined GFP::MCM-7 expression after *fos-1* RNAi. Even though *fos-1* RNAi led to a penetrant invasion defect, GFP::MCM-7 expression was not affected by *fos-1* RNAi, indicating that MCM-7 expression in the AC is independent of FOS-1 (**S6 Fig**).

In summary, the analysis of AC-specific gene expression indicated that the pre-RC positively regulates the expression of specific extracellular matrix genes that facilitate AC invasion.

## Expression profiling of human cancer cells identifies several PI3K pathway genes as candidate pre-RC targets

To systematically identify the changes in gene expression caused by inhibiting the pre-RC, we turned to human cancer cell lines. Since the pre-RC genes were originally selected based on their differential expression in human melanoma cells, we examined the changes in gene expression caused by MCM7 depletion in the A375 melanoma cell line, and to extend our study to another cancer type, included the A549 lung cancer cell line in the expression analysis.

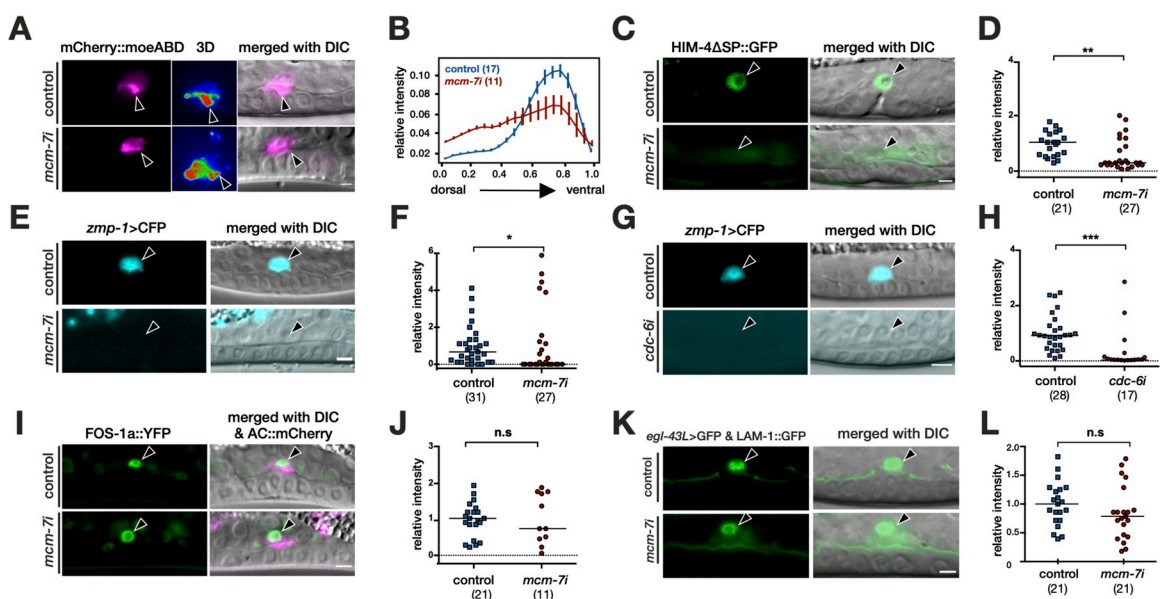

**Fig 4. The pre-RC regulates invadopodia formation and extracellular matrix gene expression in the AC.** (A) F-actin localization in the AC visualized with the mCherry::moeABD reporter (*qyIs50*) in empty vector–treated controls (top row) and after *mcm-7* RNAi (bottom row). The middle panels show single frames of 3D heat-maps obtained by reconstruction of the mCherry::moeABD signal as shown in S1 Movie. (B) Quantification of mCherry::moeABD polarity. Intensity plots along the dorso-ventral axis of the AC were generated as described in Materials and methods and in [74]. The numbers of animals analyzed are indicated in brackets. (C) *him-4>ΔSP::GFP* expression in the AC in control RNAi (top row) and in *mcm-7* RNAi-treated animals (bottom row). (D) Quantification of *him-4>ΔSP::GFP* expression levels in the AC. (E) *zmp-1>CFP* expression in the AC in control RNAi (top row) and in *mcm-7* RNAi-treated animals (bottom row). (F) Quantification of *zmp-1>CFP* expression levels in the AC. (G) *zmp-1>CFP* expression in control RNAi (top row) and in *cdc-6* RNAi-treated animals (bottom row). (H) Quantification of *zmp-1>CFP* expression levels in the AC. (I) *fos-1a>FOS-1a::YFP* expression in control RNAi (top row) and in *mcm-7* RNAi-treated animals (bottom row). (J) Quantification of *fos-1a>FOS-1a::YFP* expression levels in the AC. (K) *egl-43L>GFP* expression in control RNAi (top row) and *mcm-7* RNAi-treated animals (bottom row). (L) Quantification of *egl-43L>GFP* expression levels in the AC. For each reporter analyzed, the left panels show the fluorescence signal and the right panels the merged DIC/fluorescence images. The arrowheads point at the AC nuclei. The numbers in brackets in the graphs refer to the numbers of animals analyzed, and the horizontal lines indicate the median values. Statistical significance was determined with a Mann–Whitney test and is indicated as n.s. for $p > 0.05$, * for $p < 0.05$, ** for $p < 0.01$, and *** for $p < 0.001$. See S1 Data for the numerical values used to generate the graphs in (B), (D), (F), (H), (J), and (L). The scale bars are 5 μm. AC, anchor cell; DIC, differential interference contrast; n.s., not significant; pre-RC, pre-replication complex.

Moreover, to exclude possible effects caused by changes in cell cycle progression and focus on the replication-independent functions of MCM7, we analyzed gene expression in cells arrested at the G1/S boundary. For this purpose, we generated A375 and A549 cells stably expressing an MCM7 shRNA or a scrambled shRNA as negative control, each under control of the doxycycline (Dox)-inducible enhancer/promoter. Addition of 1 μg/ml Dox to induce shMCM7 expression resulted in a 90% to 80% reduction in MCM7 protein levels in A375 and A549 cells, respectively (**S7C–S7E Fig**). Cell cycle analysis revealed that MCM7 depletion caused a slight enrichment of cells in the G1 phase (**S7A and S7B Fig**). Using a double thymidine (Thy) synchronization protocol to arrest cells at the G1/S boundary [41], we obtained cell populations, in which over 95% of the cells were arrested in the G1 phase of the cell cycle (**S7A and S7B Fig**). From these synchronized cell populations, mRNA was extracted to obtain global expression profiles by RNAseq for both cell lines.

RNAseq analysis revealed widespread changes in the mRNA expression profile of MCM7-depleted and G1/S phase-arrested cells (**S8A and S8B Fig** and **S2 Data**). Clustering of the MCM7-regulated genes revealed an enrichment of genes encoding extracellular matrix components and cell junction and plasma membrane proteins among the up-regulated genes (**S8A and S8B Fig**). To identify a core set of MCM7 target genes, we determined the overlap

between the significantly up- and down-regulated genes in A375 and A549 cells ($\geq$1.5-fold change with $p < 0.01$ and FDR $<0.05$). This analysis identified 259 genes up-regulated and 327 genes down-regulated after MCM7 depletion in both cell lines (**S8C and S8D Fig**). GO enrichment analysis of the molecular functions and biological processes of these core MCM7 target genes indicated that many of the up-regulated genes encode extracellular matrix, integrin, actin, and ß-catenin interacting proteins, while many of the down-regulated genes encode enzymes required for mitochondrial respiration and ribosomal RNA-binding proteins (**S8E and S8F Fig**).

In particular, we observed several components of the 1-phosphatidylinositol-3-kinase (PI3K) signaling pathway among the most strongly down-regulated genes (**Fig 5A** and **S2 Data**). Expression of the regulatory PIK3R2 (p85β) and PIK3R3 (p55γ) subunits as well as the AKT3 kinase that acts downstream of PI3K was reduced upon MCM7 knock-down in G1/S-arrested A375 and A459 cells, and the PIK3CD (p110δ) catalytic PI3K subunit was also significantly reduced in A549, but not in A375 cells (**Fig 5A**). To examine whether the decreased expression of PI3K pathway components manifests in a diminished activity of the PI3K pathway, we quantified the levels of phosphorylated, activated AKT kinase using a phospho-specific AKT antibody for western blot analysis. Insulin or EGF stimulation of serum-starved A375 and A549 cells that had previously been depleted of MCM7 resulted in an attenuated activation of the PI3K pathway (**Fig 5B** and **S7C–S7E Fig**; note that A375 cells did not respond to EGF stimulation).

Moreover, we found in both cell lines significantly increased expression of TIMP2, which encodes an inhibitor of different MMPs, suggesting an involvement of MCM7 in the enzymatic BM remodeling (**S2 Data**). Mammalian cells express various MMPs, and *C. elegans zmp-1* is orthologous to several of them [7]. mRNA levels of several MMPs were significantly changed after MCM7 knock-down, but there was no overlap between the 2 cell lines. MMP9 and MMP24 expression was down-regulated and MMP3 was up-regulated in A375 cells, while MMP10 expression was up-regulated only in A549 cells (**S2 Data**).

In summary, expression profiling of 2 human cancer cell lines identified several components of the PI3K/AKT pathway and the MMP inhibitor TIMP2 as candidate MCM7 targets. Since the analysis was performed with cells arrested at the G1/S boundary, the regulation of the PI3K pathway by the pre-RC appears to occur independently of cell cycle progression.

## PI3K signaling in *C. elegans* promotes BM breaching by the AC

To examine the significance of the regulatory effect of MCM-7 on PI3K signaling, we returned to the *C. elegans* AC invasion model. In contrast to mammalians, the *C. elegans* genome encodes single catalytic (*age-1*) and regulatory (*aap-1*) PI3K subunits, respectively [42]. To observe *age-1* and *aap-1* expression, we created endogenous reporters by inserting *gfp* tags at the N-terminus of *age-1* and the C-terminus of *aap-1* using CRISPR/Cas9 genome engineering (*zh161[gfp::age-1]* and *zh178[aap-1::gfp]*). In untreated or empty vector–treated control RNAi animals, GFP::AGE-1 and AAP-1::GFP were both expressed in the AC and the surrounding cells at the Pn.pxx stage before and during invasion (**Fig 5C and 5E**). Interestingly, the GFP:: AGE-1 and AAP-1::GFP proteins were asymmetrically localized in the AC and appeared polarized toward the invasive membrane. The levels of the catalytic PI3K subunit AGE-1 decreased in the AC upon *mcm-7* RNAi (**Fig 5C–5F**). By contrast, we did not detect a significant change in the mean expression levels of the regulatory subunit AAP-1, but expression was more variable after *mcm-7* RNAi (**Fig 5F,** $p < 0.0001$ in an F-test for variance). Moreover, the asymmetric distribution of AAP-1::GFP and GFP::AGE-1 in the AC was lost in most *mcm-7* RNAi-

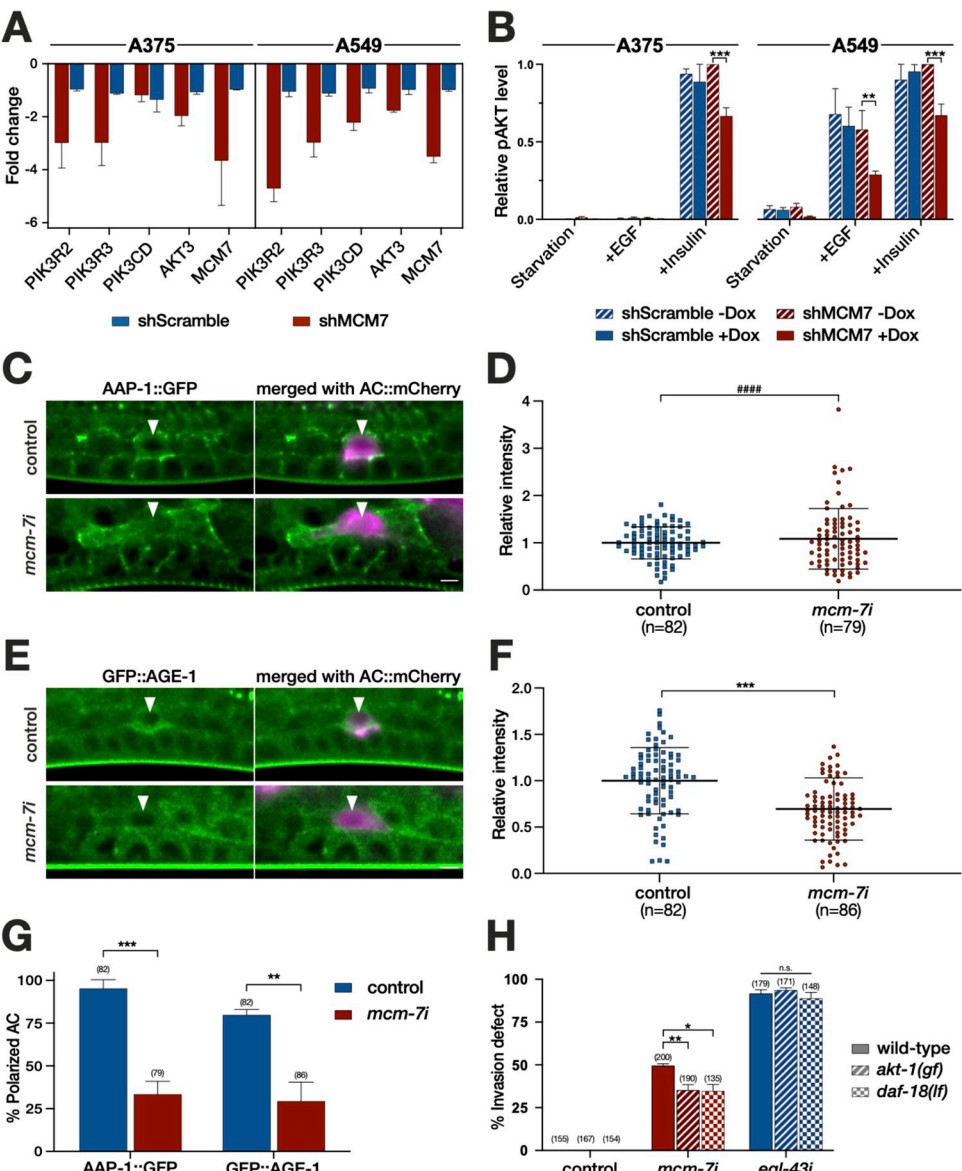

**Fig 5. The pre-RC positively regulates the PI3K pathway to promote invasion.** (A) Reduced expression of PI3K pathway genes and MCM7 after MCM7 knock-down in G1/S-arrested A375 cells (left) and A549 cells (right). For each gene, the fold change in expression between Dox-treated and untreated shMCM7 cells (red bars) and Dox-treated and untreated shScramble cells (blue bars) is shown (average of 3 biological replicates; see S2 Data). (B) Phospho-AKT protein levels in serum-starved and G1/S-arrested A375 (left) and A549 cells (right) stimulated with EGF or insulin. For each condition, pAKT levels in shMCM7 and shScramble cells grown in the presence and absence of Dox were quantified on western blots. The pAKT levels were normalized to the highest value in each experiment, and the averaged values of 3 independent biological replicates are shown (see S7C and S7D Fig). (C) Endogenous AAP-1::GFP and (E) GFP::AGE-1 expression in control (top rows) and *mcm-7* RNAi-treated *C. elegans* larvae (bottom rows) at the Pn.pxx stage. The arrowheads point to the AC labeled with the mCherry::moeABD reporter (*qyIs50*). For each reporter, the left panels show the GFP signal in green and the right panels the merged GFP (green) and mCherry (magenta) signals. (D) Quantification of AAP-1::GFP and (F) GFP::AGE-1 levels in the AC. (G) Fraction of animals exhibiting polarized AAP-1::GFP and GFP::AGE-1 localization. (H) BM breaching defects after control, *mcm-7* or *egl-43* RNAi in the indicated genetic backgrounds. The numbers in brackets in each graph refer to the numbers of animals analyzed, the horizontal lines in (D) and (F) indicate the median values and error bars the standard deviation. Statistical significance was determined by two-way ANOVA followed by two-tailed *t* tests for independent samples and is indicated as n.s. for $p > 0.5$, * for $p < 0.05$, ** for $p < 0.01$, *** for $p < 0.001$ and as **** for $p < 0.0001$. In (D), the result of an F-test for variance is indicated by #### for $p < 0.0001$. See S1 Data for the numerical values used to generate the graphs in (A), (B), (D), and (F-H). The scale bars are 5 μm. AC, anchor cell; BM, basement membrane; Dox, doxycycline; n.s., not significant; pre-RC, pre-replication complex.

treated animals (**Fig 5G**). By contrast, expression and localization of both PI3K subunits remained unchanged after *fos-1* RNAi (**S9 Fig**).

Next, we tested whether overactivation of the PI3K pathway could rescue the invasion defects caused by down-regulation of *mcm-7*. For this purpose, we performed *mcm-7* RNAi in animals carrying a gain-of-function mutation in the AKT homolog *akt-1* [43] or a deletion allele of *daf-18*, which encodes the homolog of the PTEN tumor suppressor that antagonizes PI3K activity [42], and scored BM breaching using the LAM-1::GFP marker. The *akt-1 (mg144gf)* and *daf-18(ok480lf)* mutations partially suppressed the invasion defects caused by *mcm-7* RNAi, while the *egl-43* RNAi invasion defects were not affected by *akt-1(gf)* or *daf-18 (lf)* (**Fig 5H**).

Lastly, we examined whether PI3K signaling is necessary for AC invasion by observing BM breaching in *age-1(mg44)* null mutants. The first generation (F1) of homozygous *age-1(mg44)* animals segregated by heterozygous parents developed into healthy and fertile adults, presumably due to maternal rescue of larval AGE-1 function [44]. However, their offspring (F2) exhibited a severe developmental delay and formed sterile adults with a Pvl phenotype. In 23% (*n* = 61) of the F2 generation *age-1(mg44)* mutants carrying the *lam-1::gfp* marker, no BM breaching or only a very small BM gap was observed at the L4 (Pn.pxxx) stage (**S10 Fig**), suggesting that PI3K signaling is partially required for AC invasion.

Taken together, these data suggest that the positive regulation of the PI3K pathway by the pre-RC contributes to the invasive phenotype of the AC. Since hyperactivation of PI3K signaling only partially suppressed the *mcm-7* invasion defects, the PI3K pathway is likely only one of several critical processes regulated by the pre-RC in the AC.

## Discussion

Using *C. elegans* AC invasion as an in vivo model for a targeted RNAi screen, we have made the unexpected observation that multiple components of the pre-RC positively regulate cell invasion in nonproliferating cells. To further characterize the function of the pre-RC during cell invasion, we focused on the *mcm-7* gene, because it encodes an essential subunit of the hexameric MCM complex that is the last component loaded onto the replication origins during their licensing [12,14]. After the MCM complex has been assembled, its helicase activity is activated to initiate the formation of a replication fork by unwinding the dsDNA at the replication origins. However, our analysis of *mcm-7* in the *C. elegans* AC and in human cancer cells has revealed an activity of the pre-RC that is distinct from its well-characterized role during DNA replication origin licensing. First, *mcm-7* acts cell autonomously in the postmitotic AC, which remains arrested in the G1 phase of the cell cycle. The proliferation arrest of the AC is a prerequisite for BM breaching and invasion [8]. Even though components of the pre-RC form a replication checkpoint in dividing cells [45], loss of *mcm-7* did not interfere with the G1 arrest of the AC, suggesting that the *mcm-7* invasion defects are not caused by a change in the cell cycle state of the AC. Second, the invasion defects caused by down-regulation of *mcm-7* did not correlate with and were independent of the proliferation arrest caused by loss of the pre-RC in other cells. Especially, blocking the cell cycle in the adjacent 1° VPCs that are necessary to induce AC invasion did not cause an invasion defect. Third, the DNA replication initiation complex (pre-IC) that acts downstream of the pre-RC to form a DNA replication fork is not required for AC invasion. However, an *mcm-7* rescue transgene with mutations that eliminate the ATPase activity of the MCM helicase failed to rescue the AC invasion defect, suggesting that the DNA unwinding activity of the MCM complex is needed for the transcriptional regulation of its proinvasive target genes. However, it is also possible that the MCM7 ATPase activity is needed for another unknown function of MCM7 besides its essential role in the pre-

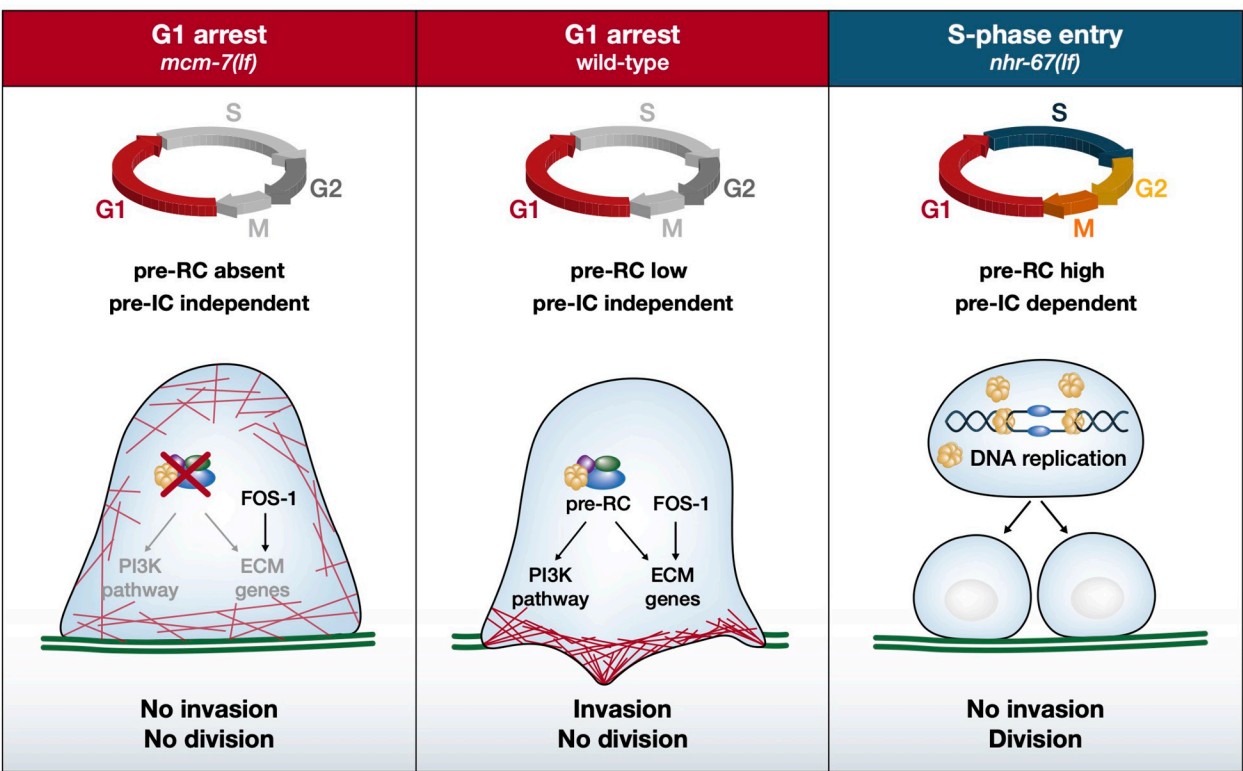

**Fig 6. Replication-independent function of the pre-RC during AC invasion.** Center: In the wild type, the AC arrests in the G1 phase of the cell cycle and expresses low levels of the MCM proteins. Under these conditions, the pre-RC controls ECM gene expression and activates the PI3K pathway, allowing the AC to polarize the actin cytoskeleton (red lines), breach the BMs (green lines), and invade. Right: If the AC is induced to proliferate, e.g., by inhibiting *nhr-67*, MCM proteins are expressed at high levels and can license the DNA replication origins. In this case, the AC does not invade but enters S-phase and divides. Left: If the pre-RC is absent, the AC remains in the G1 phase but cannot invade because ECM gene expression is deregulated and PI3K pathway activity is reduced.

RC. Forth, inhibition of *mcm-7* resulted in specific changes in the expression of extracellular matrix genes and a disorganized actin cytoskeleton at the invasive AC membrane. Yet, the inhibition of *mcm-7* function did not cause a global change in the transcription profile of the AC, as the expression of AC fate markers and the transcription factors FOS-1 and EGL-43, which are both essential regulators of AC invasion, remained unchanged [8,28,40]. Taken together, these observations strongly support a DNA replication-independent function of the pre-RC during AC invasion (**Fig 6**).

Expression profiling of 2 human cancer cell lines arrested at the G1/S-phase boundary identified several PI3K pathway components as candidate pre-RC targets. Many PI3K pathway genes are involved in the malignant progression of various cancer types [46]. Especially, increased expression of the regulatory subunit PIK3R2 has consistently been associated with an advanced tumor stage and enhanced metastasis formation [47]. Moreover, a link between MCM7 and the PI3K pathway has previously been reported for human melanoma [48] and esophagal squamous carcinoma [49], and an inhibitory role of the MCM7 locus on the PI3K antagonist PTEN was observed in a transgenic mouse model [50]. Likewise, depletion of *mcm-7* in *C. elegans* also lead to reduced expression of the catalytic PI3K subunit *age-1*, while increasing PI3K pathway activity partially suppressed the AC invasion defect caused by *mcm-7* depletion. The PI3K pathway may be activated in the AC by the INA-1/PAT-3 αβ-integrin complex on the invasive membrane [6] and signal via MIG-10/Lamellipodin to promote

invasion, analogous to the interaction between AGE-1 and MIG-10 during axon guidance [51]. Accordingly, a loss of *age-1* function caused partially penetrant BM breaching defects.

A replication-independent role of the pre-RC in the transcriptional regulation of specific targets has first been observed in budding yeast during the silencing of mating-type genes [52,53]. Also, the mammalian MCM proteins negatively regulate the transcriptional activity of the hypoxia inducible factor HIF-1α in quiescent cells [54], while MCM5 is required for Stat1--mediated transcriptional activation [55]. The genome-wide mapping of replication origins in *C. elegans* and in human cells revealed an enrichment of active, "firing" origins near transcription factor binding sites in the vicinity of highly transcribed genes [56,57]. It thus appears that the pre-RC has adopted a new function in the postmitotic AC to license the expression of proinvasive genes (**Fig 6**). It should be noted that even a global inactivation of *mcm-7* by FLP/FRT-mediated excision only caused a partially penetrant BM breaching defect. Hence, other factors may compensate for a loss of pre-RC activity to facilitate the switch toward an invasive fate.

Interestingly, the expression levels of MCM-7 in the AC of L3 larvae were lower than in the adjacent uterine and vulval cells that continued to proliferate, and ectopic expression of *mcm-7* using a strong AC-specific enhancer element led to relatively low MCM-7 protein levels, suggesting that MCM-7 is down-regulated in the G1-arrested AC at the posttranscriptional level. Dividing cells usually express an excess of MCM proteins that assemble not only on active replication origins but are spread over the entire chromatin at sites that do not overlap with sites of DNA replication, a phenomenon termed "MCM paradox" [14]. It has been proposed that the excess of MCM complexes in dividing cells are used to license a large number of dormant origins that could be used as reserve in case of DNA replication stress [58]. The relatively low MCM-7 levels in the G1-arrested AC may confer a certain specificity to the MCM complex by allowing it to bind only to a limited number of high-affinity sites near its target genes, while being insufficient to permit the progression through S-phase.

Several components of the pre-RC are overexpressed in tumor cells and have therefore been used as biomarkers [13,14]. The pre-RC has been predominantly studied in the context of cell cycle progression and genomic instability in human cancer cells. However, there exists a handful of reports associating pre-RC genes with cancer progression and increased cell migration [49,59–62]. Notably, Lau and colleagues observed reduced migration and invasion of human medulloblastoma cells after inhibition of MCM proteins [63]. Our data using a developmental model for cell invasion provide direct evidence for a cell cycle–independent function of the pre-RC during cell invasion. For some pre-RC components, DNA replication-independent functions that involve distinct protein complexes have been reported: For example, *Drosophila* ORC6 forms a complex with the septin protein Peanut to drive cytokinesis [64], and CDT1 has been proposed to act as a hub for the recruitment of DNA repair proteins [65] and chromatin modifiers such as HBO1 [66], HDAC11 [67], and Geminin [68]. Thus, it is also possible that the pre-RC components we have identified in this study act outside of the canonical pre-RC complex to perform their proinvasive function by forming complexes with other cellular proteins.

In summary, the proinvasive function of the pre-RC—or some of its components—may be conserved between *C. elegans* and mammalian cells. The changes in extracellular matrix and PI3K gene expression after down-regulating *mcm-7* in the *C. elegans* AC and in human cancer cells suggest that the pre-RC controls cell adhesion and BM remodeling to facilitate cell invasion. Likewise, cells entering S-phase turn over their focal adhesions, while mitotic cells must completely detach from the matrix to undergo cytokinesis [69]. Therefore, the pre-RC may coordinate cell adhesion with DNA replication in dividing cells, while invasive cells such as the

AC may have uncoupled the regulation of cell adhesion and BM breaching by the pre-RC from its function in replication origin licensing.

## Materials and methods

### *C. elegans* culture and maintenance

*C. elegans* strains were maintained at 20˚C on standard nematode growth plates as described [70]. The wild-type strain was *C. elegans* Bristol, variety N2. We refer to translational protein fusions with a:: symbol between the gene name and the tag, while transcriptional fusions are indicated with a > symbol between the promoter/enhancer and the tag. The genotypes of the strains used in this study are listed in **S3 Table**. Except for strain PS4444, which was provided by CGC, all strains were generated in this study.

### Scoring AC invasion

AC invasion was scored at the Pn.pxx (4-cell) stage as described [8]. We monitored the continuity of the BM by fluorescence microscopy using the *qyIs10[lam-1>lam-1::gfp]* or *qyIs127 [lam-1>lam-1::mCherry]* transgenes as markers.

### Global and tissue-specific *RNA interference*

RNAi by feeding dsRNA-producing *E. coli* was performed in strains carrying the indicated markers in the *rrf-3(pk1426)* RNAi hypersensitive background as described in [71,72]. Synchronized populations of L1 larvae were obtained by hypochlorite treatment of gravid adults. If the RNAi treatment did not cause lethality or sterility, then the F1 generation was scored after 5 days, or else the P0 animals were analyzed after 30 to 36 hours of treatment. RNAi clones targeting genes of interest were obtained from the *C. elegans* genome-wide RNAi library [71] or the *C. elegans* open reading frame (ORFeome) RNAi library (both from Source BioScience). Genes not available in either of the 2 libraries were cloned into the L4440 vector by Gibson assembly using the oligonucleotide primers indicated in **S4 Table**. For AC- and Pn.p cell-specific RNAi, we used an *rde-1(ne219); rrf-3(pk1426); qyIs10[lam-1>lam-1::gfp]* strain carrying the *qyIs102[fos-1a>rde-1]* or the *zhEx418[lin-31>rde-1]* transgene, respectively [31,32]. The empty L4440 vector was used as negative control and an RNAi clone targeting *egl-43* [40] was used as positive control.

### Microscopy of *C. elegans*

Fluorescent and Nomarski images of *C. elegans* larvae were acquired with a LEICA DM6000B microscope equipped with a Leica DFC360 FX camera and a 63x (N.A. 1.32) oil-immersion lens, or with an Olympus BX61 wide-field microscope equipped with a X-light spinning disc confocal system using a 70-μm pinhole, a lumencor solid-state light source using a 60x Plan Apo (N.A. 1.4) lens and an iXon Ultra 888 EMCCD camera.

### AC polarity and fluorescence intensity measurements

Fluorescence intensities of the different reporters were quantified using Fiji software with the built-in z-projection and measurement tools [73]. To measure the intensity of the different reporters, background subtracted summed z-projections spanning the width of the AC labeled with an mCherry marker were quantified. Quantification of mCherry::moeABD polarity in the AC was done using deconvolved wide-field or confocal image stacks spanning the width of the AC. Polarity plots were calculated from the summed z-projections of the AC as described in [74]. Quantification of the CKD sensor was done according to [39]; a mid-sagital section of

the AC was filtered with Gaussian blur (sigma = 50) to reduce camera noise, and the average intensities in three randomly selected areas, each in the AC nucleus and cytoplasm, were used to calculate the cytoplasmic to nuclear intensity ratios.

### Generation of reporter and rescue transgenes

For all constructs, fragments were PCRs amplified with Phusion DNA Polymerase (New England Biolabs) and were assembled by T4 ligase (New England Biolabs) or if Gibson assembly was used with Gibson assembly kit #E2611 (New England Biolabs). Details on the construction of the plasmids used to create the different transgenes are shown in **S5 Table**. The sequences of the oligonucleotide primers used can be found in **S4 Table**. All plasmids were verified by DNA sequencing. For each reporter, single-copy transgene insertions were created according to the MosSCI protocol [37] by microinjection of 50 ng/µl of the respective reporter plasmid together with 50 ng/µl pJL43.1 (*mos-1* transposase) and 2.5 ng/µl pCFJ90 (*myo-2>mCherry*), 5 ng/µl pcFJ104 (*myo-3>mCherry*), and 10 ng/µl pGH8 (*rap-3>mCherry*) as coinjection markers.

### Generation of the zh118[frt::gfp::mcm-7], zh161[gfp::age-1], and zh178 [aap-1::gfp] alleles

The streamlined CRISPR/CAS9 method described [25] was used to generate endogenously tagged reporters. The 5′ and 3′ homology arm were amplified from N2 genomic DNA with primers indicated in **S5 Table**. These fragments were cloned into pMW75 digested with *Cla*I to create the repair templates pTD16, pTD60, and pTD72. pMW75 is a derivate of pDD282, in which the *gfp* cassette and the ccdB negative selections markers were replaced with a *gfp* cassette from pDD95.75 containing FRT-sites in introns 1 and 2. About 20 ng/µl of the resulting plasmids were injected together with 50 ng/µl of the respective single-guide RNA expression plasmids pTD17 (*mcm-7* sgRNA#2 (CCACCACTTACAACACAGAC), pTD61 (*age-1* sgRNA#1 TTA AGA TTT TAA GAT GTC TA) and pTD62 (*age-1* sgRNA#2 TAA GAT TTT AAG ATG TCT AT) and pTD63 (*age-1* sgRNA#3 GGT TCG ACT TCG AAA TGT CG) or pTD73 (*aap-1* sgRNA#1 AAC TAT TCC GGG ATG GAT TT) and pTD74 (*aap-1* sgRNA#2 TTT AAA TAT ATA ACT ATT CC) and 2.5 ng/µl pCFJ90 (*myo-2>mCherry*), 5 ng/µl pcFJ104 (*myo-3>mCherry*), and 10 ng/µl pGH8 (*rap-3>mCherry*) coinjection markers. The selection of the integrated CRISPR lines was carried out according to [25].

### FLP/FRT-mediated mcm-7 mosaic analysis and AC-specific rescue

Mosaic analysis in the *qyIs10[lam-1>lam-1::gfp]; qyIs23[cdh-3>mCherry::PLCδPH]; zh118 [frt::gfp::mcm-7]* strain was performed by FLPase expression from a heat shock–inducible promoter, using the *zhEx621[hs-16-48>flp]* transgene as described [26]. Late L1/early L2 larvae were heat shocked for 1 hour at 34˚C and allowed to recover at 20˚C. GFP::MCM-7 expression and AC invasion was scored 40 hours later. For AC-specific rescue, the wild-type *zhIs119* and mutant *zhIs142* transgenes were introduced into the *zhEx621[hs-16-48>flp]; qyIs10[lam-1>lam-1::gfp]; qyIs23[cdh-3>mCherry::PLCδPH]; zh118 [frt::gfp::mcm-7]* background and BM breaching was scored after heat shock induced *flp* expression, as described above.

### Generation of Dox-inducible shMCM7 and shControl cells

The human cell lines A375 and A549 were tested to be free of mycoplasm and cultured in Dulbecco's Modified Eagle Medium (DMEM, Gibco 41966–029) supplemented with 10% FCS

(Gibco 10500–064) and 1% Pen-Strep (Gibco 15140–122) according to standard mammalian tissue culture protocols and sterile technique.

Double stranded oligonucleotides for an shRNA targeting MCM7 (CAG AGG AGG ATT TCT ACG AAA), or a scrambled shRNA (TAA TGT ATT GGA ACG CAT ATT) were cloned into the Dox-inducible shRNA expression vector pRSIT17 (Cellecta # SVSHU6T17-L), which was then transfected in combination with pVSV-G, pMDL and pREV into HEK293T cells to produce lentiviral particles. Four days following transfection, the media from cells was collected, clarified by centrifugation, and filtered through a 0.45-μM filter to collect lentiviral particles. Subsequently, the particles were concentrated in Amicon Ultra tubes (Ultracell 100k, Millipore). A375 or A549 cells were transduced overnight with lentiviral particles (MOI of 1.4) in DMEM medium supplemented with 10 μg/ml polybrene. Three days after transduction, cells were grown in the presence of 1.2 μg/ml puromycin. One week after puromycin selection, the puromycin-resistant populations were frozen and kept as stocks used for the subsequent experiments.

## Cell synchronization and cell cycle analysis

Cell synchronization with double Thy block [41] was performed by growing cells in medium supplemented with 2 mM Thy for 17 hours, followed by an 8-hour release in medium without Thy, after which cells were kept arrested in medium containing 2 mM Thy during the analysis. Cell cycle analysis by DAPI and EdU double labeling was done using the Click-iT assay (Thermo Fischer C10420) as described [75] and analyzed by flow cytometry (BD LSR II Fortessa).

## RNAseq analysis of Dox-inducible shMCM7cells

Total RNA from Dox-inducible shMCM7 or parental A375 and A549 cells, each arrested in G1 by double Thy block and simultaneously treated for 72 hours with 1 μg/ml Dox, where indicated, was isolated using the RNeasy Mini Kit (#74104 Qiagen). Three biological replicates for each condition were analyzed. Quality control by Agilent Tape Station/Bioanalyzer, RNA library preparation, and whole exome sequencing were performed in triplicates by the transcriptomics service of the Functional Genomics Center Zürich (FGCZ). Using the HiSeq 2500 System (Illumina), around 200 million reads/lane were obtained. Bioinformatic analysis was done by the FGCZ using an in-house pipeline utilizing the open source tools STAR for read alignment, DESeq2 and EdgeR for differential gene expression, and GOseq for GO enrichment analysis. Bootstrap analysis with a resampling size of 1,000 was used to determine the significance of the overlap between the MCM7-regulated genes in A549 and A375 cells. The RNAseq data generated in this study are available at the NCBI Gene Expression Omnibus (GEO) (http://www.ncbi.nlm.nih.gov/geo/) under accession number **GSE149523**.

## Western blot analysis

Cells were lysed in lysis buffer on ice (100 mM Tris/HCl, 150 mM NaCl, 1% Triton X, 1 mM EDTA, 1 mM DTT, 10 ml lysis buffer + 1 tablet protease inhibitor), scraped with a cell scraper, and snap frozen in liquid nitrogen. A volume of 100 μl of this mix was sonicated in a Bioruptor sonicator device (Diagenode), before 100 μl of 2x SDS loading dye were added. About 10 mg of protein extract were resolved by SDS-PAGE and transferred to nitrocellulose membrane. The membrane was blocked in 5% dried milk in 1x PBS plus 0.2% Tween 20 and incubated with the diluted primary antibodies overnight at 4°C. Secondary anti-rabbit or anti-mouse IgG antibodies conjugated to horseradish peroxidase (HRP) were used as the secondary antibodies. The HRP was detected by incubating the membrane with the SuperSignal West Pico or

Dura Chemiluminescent Substrate (Thermo Scientific) for 4 minutes, before the signals were acquired on a digital chemiluminescence imaging system. The antibodies used for western blot analysis were anti-MCM7 (141.2 Santa Cruz), anti-Tubulin (ab18251 abcam), and anti-pAKT (44-621G Thermo Scientific). Quantification of western blots was done by measuring the band intensities in Fiji.

### EGF and insulin stimulation

About 28,000 cells were seeded into 12-well plates and treated after 24 hours throughout the experiment with 1 μg/ml Dox where indicated. Around 48 hours after seeding, cells were additionally arrested at the G1/S boundary by growing cells in medium supplemented with 2 mM Thy for 17 hours, followed by an 8-hour release in medium without Thy [41]. Thereafter, growth medium was replaced with DMEM containing 2 mM Thy and lacking FCS. After 15 hours of starvation, cells were stimulated for 10 minutes with 100 ng/ml human EGF (E9644 Sigma) or with 10 nM human insulin (I9278 Sigma) before they were lysed and prepared for western blot analysis.

### Supporting information

**S1 Fig. Structure of the *frt::gfp::mcm-7(zh118)* locus.** Structure of the *frt::gfp::mcm-7(zh118)* locus before and after FLP-mediated recombination. The *gfp* coding regions and protein sequence are labeled in green and the *mcm-7* region in yellow. The stop codons generated by the frameshift after the excision of exon 2 are indicated in red. FLP, flippase; FRT, FLP recognition target sequence.
(PDF)

**S2 Fig. *Cdt-1* is expressed in the AC during invasion.** GFP::CDT-1 (*zhIs120*) and LAM-1::GFP expression beginning at the mid-L2 stage, shortly before AC specification, until the late-L3 (P6.pxx) stage after BM breaching are shown. DIC images are shown in the left panels; CDT-1::GFP and LAM-1::GFP expression in green are shown in the middle panels. The right panels show the GFP images merged with the AC marker *ACEL>mCherry* (*zhIs127*) in magenta. The arrowheads point to the AC nuclei. The scale bar is 5 μm. AC, anchor cell; BM, basement membrane; DIC, differential interference contrast.
(PDF)

**S3 Fig. Expression of AC and VPC fate markers after *mcm-7* RNAi.** (**A**) Expression of a translational reporter for the HLH-2 transcription factor together with the LAM-1::GFP BM marker in control animals (top row) and in *mcm-7* RNAi-treated animals (bottom row) and (**B**) quantification of HLH-2 expression levels in the AC. (**C**) Expression of a transcriptional reporter for the CDH-3 proto-cadherin together with the LAM-1::GFP BM marker in control animals (top row) and in *mcm-7* RNAi-treated animals (bottom row) and (**D**) quantification of *cdh-3* expression levels in the AC. (**E**) Expression of a transcriptional reporter for the EGF homolog *lin-3* in control animals (top row) and in *mcm-7* RNAi-treated animals (bottom row) and (**F**) quantification of *lin-3* expression levels in the AC. (**G**) Expression of the 1° fate marker *egl-17* in the VPCs of control animals (top row) and in *mcm-7* RNAi-treated animals (bottom row). Left panels show the GFP signals and right panels the GFP channel merged with the DIC images. The numbers in brackets in the graphs refer to the numbers of animals analyzed, and the horizontal lines indicate the median values. Statistical significance was determined with a two-tailed *t* test for independent samples and is indicated as n.s. for $p > 0.05$ and *** for $p < 0.001$. See **S1 Data** for the numerical values used to generate the graphs in (**B**), (**F**), and (**D**). The scale bars are 10 μm in (**A**) and 5 μm in (**C**), (**E**), and (**G**). AC, anchor cell; DIC,

differential interference contrast; n.s., not significant; VPC, vulval precursor cell.
(PDF)

**S4 Fig. AC-specific expression of wild-type and helicase-deficient MCM-7::CFP.** (A) Alignment of the ATPase motif in the helicase domain of human and *C. elegans* MCM-7. Conserved residues are highlighted in black. The 4 amino acid substitutions eliminating the helicase activity in the *ACEL>mcm-7(mut)::cfp* transgene are shown. (B) AC-specific expression of the *ACEL>*MCM-7::CFP wild-type and mutant proteins. The arrowheads in the DIC images point at the AC nuclei. (C) Quantification of ACEL>MCM-7(wt)::CFP expression in the somatic gonad from the L1 to the ate L2 stage. Larval stages were assigned according to the gonad lengths, as described in [76]. The x-axis shows the gonad length measured as distance between the 2 distal tip cells and the y-axis the proportion of animals showing MCM-7(wt):: CFP expression in the AC (*n* = 391). See S1 Data for the numerical values used to generate the graph. The scale bar in (B) is 5 μm. AC, anchor cell; DIC, differential interference contrast. (PDF)

**S5 Fig. *hda-1* expression after *mcm-7* RNAi.** (A) HDA-1::RFP expression in the AC in control RNAi (top row) and *mcm-7* RNAi-treated animals (bottom row). The scale bar is 5 μm. (B) Quantification of HDA-1::RFP expression levels in the AC. The right panels show the DIC images and the right panels the fluorescence signal. The arrowheads point at the AC nuclei. The numbers in brackets in the graphs refer to the numbers of animals analyzed, and the horizontal lines indicate the median values. Statistical significance was determined with a two-tailed *t* test for independent samples and is indicated as *** for $p < 0.001$. See S1 Data for the numerical values used to generate the graph. AC, anchor cell; DIC, differential interference contrast. (PDF)

**S6 Fig. GFP::MCM-7 expression after *fos-1* RNAi.** (A) GFP::MCM-7 expression in the AC in control RNAi (top row) and *fos-1* RNAi-treated animals (bottom row) The scale bar is 5 μm. (B) Quantification of GFP::MCM-7 expression levels in the AC. The right panels show the DIC images overlaid with the GFP::MCM-7 and the AC::mCherry signals. The numbers in brackets in the graphs refer to the numbers of animals analyzed, and the horizontal lines indicate the median values. See S1 Data for the numerical values used to generate the graph. AC, anchor cell; DIC, differential interference contrast. (PDF)

**S7 Fig. Knock-down of MCM7 in G1/S-arrested A375 and A549 cells.** (**A**) Cell cycle analysis by flow cytometry of A375 and (**B**) A549 cells expressing shScramble or shMCM7 without or with double thymidine block (+Thy). (**C**) Western blot analysis of phospho-AKT and MCM7 levels upon MCM7 knock-down in Thy-arrested, serum-starved EGF and insulin-stimulated A375 and (**D**) A549 cells. The western blots of 1 out of 3 biological replicates for each cell line are shown. See **S1 Raw Images** for the original images of all western blots. (**E**) Quantification of MCM7 protein levels in shScramble and shMCM7 expressing A375 and A549 cells. The averaged measurements of 4 biological replicates are shown. Statistical significance in (**E**) was determined with two-way ANOVA followed by two-tailed *t* tests for independent samples and is indicated as * for $p < 0.05$ and as **** for $p < 0.0001$. See **S1 Data** for the numerical values used to generate the graph s in (**A**), (**B**), and (**E**). The flow cytometry data can be found in the FlowRepository (https://flowrepository.org/) under accession number **FR-FCM-Z4XE**. Dox, doxycycline; Thy, thymidine. (PDF)

**S8 Fig. Expression profiles of MCM7-depleted and G1/S-arrested A549 and A375 cells.** (A) Differentially expressed genes after Dox-inducible MCM7 knock-down in G1/S-arrested (+Thy) A375 cells and (B) A549 cells were identified by RNAseq analysis. For each cell line, the data of 3 independent biological replicates were analyzed. A clustering analysis of up- and down-regulated genes with a >1.5-fold expression change with $p < 0.01$ and an FDR <0.05 is shown. Control cells were the parental A549 and A375 cell lines grown with or without Dox. For all samples, RNA was extracted from populations, in which at least 95% of the cells had been arrested in the G1/S phase boundary by double Thy block as illustrated in S7 Fig. The colored boxes on the y-axes indicate the identified gene clusters according to the cellular compartments of their gene products. The significantly changed transcripts are listed in S2 Data, and the RNAseq data generated in this study are available at the NCBI GEO (http://www.ncbi.nlm.nih.gov/geo/) under accession number GSE149523. (C) Overlap between the up-regulated and (D) down-regulated genes in the 2 cell lines. The overlap between the MCM7-regulated genes in the 2 cell lines is highly significant ($p < 0.001$), as the expected number of randomly overlapping genes determined by bootstrap analysis is 42 for the up- and 52 for the down-regulated genes. (E) GO enrichment analysis according to the biological processes and (F) molecular functions of the genes regulated by MCM7 in both cell lines. The red or orange bars indicate the top 10 GO terms enriched in up-regulated genes and the dark or light blue bars the top 10 GO terms enriched in down-regulated genes. The lengths of the bars correspond to the ±log10 of the $p$-values indicating the significances of enrichment in each cell line. Dox, doxycycline; FDR, false discovery rate; GEO, Gene Expression Omnibus; GO, Gene Ontology; Thy, thymidine.
(PDF)

**S9 Fig. AAP-1::GFP and GFP::AGE-1 expression after *fos-1* RNAi. (A)** Quantification of AAP-1::GFP and **(B)** AGE-1::GFP expression levels in the AC of control and *fos-1* RNAi-treated animals. **(C)** Fraction of animals exhibiting polarized AAP-1::GFP and GFP::AGE-1 localization. The numbers of animals scored are indicated in brackets. The horizontal lines indicate the median values. No significant differences were found in two-tailed *t* tests for independent samples ($p > 0.2$). See **S1 Data** for the numerical values used to generate the graphs in (**A-C**). AC, anchor cell.
(PDF)

**S10 Fig. BM breaching in *age-1 PI3K* null mutants.** (**A**) BM breaching visualized with the LAM-1::GFP marker in *age-1(mg)/+* heterozygous and (**B**) homozygous *age-1(mg44)* L4 larvae in the F2 generation. All heterozygous *age-1(mg)/+* siblings carrying the *mnC1* balancer showed normal BM breaching ($n = 38$), while 23% of the F2 generation *age-1(mg44)* animals ($n = 61$) exhibited no or only partial BM breaching. BM, basement membrane.
(PDF)

**S1 Table. RNAi screen of *C. elegans* homologs of human genes differentially expressed in invasive cells.** List of human genes (first column) that changed their expression with a log2 fold change ≥1 after SOX9 overexpression, TGF-ß stimulation, under hypoxia or during NC cell migration as indicated by a +. The names of the *C. elegans* orthologs that were screened by RNAi for AC invasion defects are indicated in the sixth and seventh column. Pvl indicates a protruding vulva phenotype at the adult stage, which is characteristic of vulval morphogenesis or AC invasion defects. The frequencies of BM breaching defects or of delayed BM breaching (after the Pn.pxx stage) are indicated. RNAi of genes causing a BM breaching defect are highlighted in green, and the negative control RNAi with the empty vector in red. n refers to

the number of animals scored. AC, anchor cell; BM, basement membrane; NC, neural crest.
(XLSX)

**S2 Table. The pre-IC is not required for AC invasion.** Genes encoding different pre-IC components were analyzed by RNAi for their potential to breach the BM at the P6.p.xx stage as described in Fig 2. Observed BM breaching defects are indicated in absolute numbers. AC, anchor cell; BM, basement membrane; pre-IC, pre-initiation complex.
(XLSX)

**S3 Table. List of *C. elegans* strains used.**
(XLSX)

**S4 Table. List of oligonucleotide primers used.**
(XLSX)

**S5 Table. Construction of plasmids used to generate reporter lines.**
(XLSX)

**S1 Data. Numerical values used to generate the graphs, as indicated in the figures.**
(XLSX)

**S2 Data. Results of the RNAseq analysis in A375 and A549 cells.** Genes with a >1.5-fold expression change with $p < 0.01$ and an FDR <0.05 are included. The data for each individual cell line as well as the overlap between the 2 lines are shown. FDR, false discovery rate.
(XLSX)

**S1 Raw images. Uncropped images of the western blots used for the quantifications shown in Figs 5B and S7.**
(PDF)

**S1 Movie. 3D heat-maps of moeABD expression in the AC.** 3D reconstructions of the mCherry::moeABD signal in the AC after control (left) and *mcm-7* RNAi (right). AC, anchor cell.
(AVI)

## Acknowledgments

We wish to thank the members of the Hajnal laboratory for critical discussion and comments on the manuscript. We thank Michelle Gut for experimental support, the Functional Genomic Center Zurich FGCZ, particularly Susanne Kreutzer and Giancarlo Russo, for their assistance with the RNAseq analysis, Tinri Aegerter-Wilmsen for assistance with data analysis, Franziska Walser for her assistance and support, and the Flow Cytometry Facility UZH for performing flow cytometry analyses. We are also grateful to the *C. elegans* Genetics Center CGC, which is funded by NIH Office of Research Infrastructure Programs (P40 OD010440), and the Mitani lab (National Bioresource Project) for providing some strains, the van der Heuvel lab for SV1667 strain, Andrew Fire for GFP vectors, the Erik Jorgensen Lab for plasmid pWD79-2RV, J. Ahringer for RNAi clones, and the Greber Lab for A549 cells.

## Author Contributions

**Conceptualization:** Evelyn Lattmann, Ting Deng, Michael Walser, Reinhard Dummer, Mitchell P. Levesque, Alex Hajnal.

**Data curation:** Ossia Eichhoff.

**Formal analysis:** Evelyn Lattmann, Ting Deng, Michael Walser, Alex Hajnal.

**Funding acquisition:** Reinhard Dummer, Mitchell P. Levesque, Alex Hajnal.

**Investigation:** Evelyn Lattmann, Ting Deng, Michael Walser, Patrizia Widmer, Charlotte Rexha-Lambert, Vibhu Prasad, Michael Daube.

**Methodology:** Patrizia Widmer, Charlotte Rexha-Lambert, Vibhu Prasad, Ossia Eichhoff, Michael Daube.

**Project administration:** Alex Hajnal.

**Resources:** Ossia Eichhoff, Michael Daube.

**Supervision:** Evelyn Lattmann, Reinhard Dummer, Mitchell P. Levesque, Alex Hajnal.

**Visualization:** Evelyn Lattmann, Ting Deng, Michael Walser, Alex Hajnal.

**Writing – original draft:** Evelyn Lattmann, Ting Deng, Michael Walser.

**Writing – review & editing:** Alex Hajnal.

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
