## [Editor Report · Decision Letter 0]

8 Jun 2021

Dear Dr Hajnal, 

Thank you for submitting your manuscript entitled "A DNA Replication-Independent Function of the pre-Replication Complex during Cell Invasion in C. elegans" for consideration as a Research Article by PLOS Biology.

Your manuscript has now been evaluated by the PLOS Biology editorial staff as well as by an academic editor with relevant expertise and I am writing to let you know that we would like to send your submission out for external peer review.

Please re-submit your manuscript within two working days, i.e. by Jun 10 2021 11:59PM.

Kind regards,

Richard Hodge, PhD

Associate Editor

PLOS Biology

rhodge@plos.org

---

## [Decision Letter · Decision Letter 1]

1 Jul 2021

Dear Dr Hajnal,

Thank you very much for submitting your manuscript entitled "A DNA Replication-Independent Function of the pre-Replication Complex during Cell Invasion in C. elegans" for consideration as a Research Article at PLOS Biology. Your manuscript has been evaluated by the PLOS Biology editors, an Academic Editor with relevant expertise, and by two independent reviewers.

You will see that both reviewers agree that the role for the pre-RC in promoting cell invasion is an exciting finding of broad interest, but also think that several revisions should be done before we can consider the manuscript for publication. Reviewer 1 would like you to check the timing of the genetic manipulations to show clearly at what point they affect the invading cell lineage, and wants an additional RNAi experiment to establish whether PI3K signalling is required for invasion on its own. Reviewer 2 raises some alternative interpretations of the results, in particular that the invasion phenotype could be mediated by pre-RC components interacting with other multi protein complexes. After discussing the reviews with the Academic Editor and the rest of the team, we do think these experiments should be performed.

In light of the reviews (attached below), we will not be able to accept the current version of the manuscript, but we would welcome re-submission of a revised version that takes into account the reviewers' comments. We cannot make any decision about publication until we have seen the revised manuscript and your response to the reviewers' comments. Your revised manuscript is also likely to be sent for further evaluation by the reviewers.

We expect to receive your revised manuscript within 3 months. 

**IMPORTANT - SUBMITTING YOUR REVISION**

3. Resubmission Checklist

a) *Published Peer Review*

b) *PLOS Data Policy*

Please also make sure you mention in the corresponding figure legends WHERE THE DATA CAN BE FOUND, and that your Data Statement in the submission system accurately describes where your data can be found.

Sincerely,

Ines Alvarez-Garcia, PhD

Senior Editor

PLOS Biology

on behalf of

Richard Hodge

Associate Editor

PLOS Biology

rhodge@plos.org

Reviewers’ comments

Rev. 1:

In the submitted work, "A DNA-Replication-Independent Function of the pre-Replication Complex during Cell Invasion in C. elegans" the authors present evidence for a new role for the pre-Replication Complex (pre-RC) during anchor cell invasion in C. elegans. Using an RNAi screen based on genes upregulated in human melanoma cells, they show that a number of components of the pre-RC are required for AC (anchor cell, the invading cell) invasion (CDC-6, ORC-2, ORC-5, CDT-1, MCM-7). They further demonstrate using AC-specific RNAi and AC-specific rescue that MCM-7, a component of the pre-RC, functions in the AC after its specification to promote cell invasion through basement membranes (and that it requires its enzymatic helicase activity—suggesting a role in modifying DNA). Further, they provide RNAi knockdown evidence, that components of the preinitiation Complex (pre-IC) that act downstream of pre-RC during S phase are not required for AC invasion. They go onto endogenously tag MCM-7 and show that it is expressed at low levels in the post-mitotic AC (but then is not detectable just prior to and during invasion). The authors show that MCM-7 does not alter the fate nor the cell cycle status of the AC, but instead promotes the expression of several pro-invasive genes (the MMP, ZMP-1 and the matrix protein hemicentin), as well as regulates the tight localization of F-actin at the invasive front during BM breaching. The authors go back to melanoma cells and identify the PI3K signaling pathway as a potential positively regulated transcriptional target of MCM7 in two melanoma lines and go onto show that PI3K is expressed in the AC (AGE-1 and AAP-1) and that the protein localization (and possibly gene expression for age-1) is regulated by the C. elegans MCM-7. Finally, the authors show that mutations that activate PI3K signaling in the AC partially rescue the mcm-7 RNAi mediated defect in AC invasion.

Overall, this is a well written, rigorous, and important and interesting study that uses the strengths of C. elegans to reveal an unexpected role for the pre-RC in promoting cell invasion—a finding that is important for expanding our understanding of the roles of pre-RC, as well as our understanding of cell invasion in development and cancer. I have few comments, however, that should be addressed to provide more rigor, accuracy and depth to the study.

Points to address:

1. One of the most important and interesting points of the paper is that the pre-Replication complex has a role in the post-mitotic AC to promote cell invasion. Because the lineage of the AC is complex and the timing of gene loss for most experiments in the work is not precise and likely before the AC is specified, this point needs to be clarified in the manuscript (esp., since the pre-RC has an established role in DNA replication associated with cell division). To reduce the function of mcm-7 and implicate a role in the post mitotic AC, the authors in part use a FLP/FRT-induced mosaic knockout system for mcm-7 that is induced via heat shock at the late L1/early L2 stage. One of the points the authors should state in this section of the results is that the AC is born in the early L2 stage (either Z1.ppp or Z4.aaa). Any mcm-7 knockdown before that time (such as possibly with the FLP/FRT system given the timing of induced knockout, and with whole body RNAi, which most of the work is done with) could implicate a role for mcm-7 in the mother cell that gave rise to the AC or in the cell division that gave rise to the AC rather than a role in the post-mitotic AC. This section should be organized to emphasize how the AC-specific RNAi establishes a role for mcm-7 in the post-mitotic AC itself (including describing specifically when the AC-specific strain becomes sensitive to RNAi) and not the ACs predecessor/mother cell. This point should also be emphasized in the following section with the control rescue of the mcm-7 knockout when mcm-7::cfp is driven by the lin-3 AC-specific promoter. To me this is the most important experiment in the paper, and I would have preferred an AC-specific degradation approach for MCM-7 (ZIF or AID), however, the lin-3 ACEL driven rescue is convincing, as long as early expression driven from this regulatory region is ruled out.

2. The authors should confirm when mcm-7::CFP is first seen in the AC when driven by the AC-specific LIN-3 element—especially to make certain it is not expressed in the L1 stage in the gonad and cell that gives rise to the AC.

3. The authors state, "Invasion by the post-mitotic AC is linked to the differentiation and proliferation of the underlying 1° VPCs that produce guidance signals, which attract the AC ventrally [7]. We therefore tested whether the AC invasion defects observed after down-regulation of pre-RC components are caused by a proliferation arrest of the VPCs." Notably, it has already been demonstrated that the AC can invade when the VPC divisions are blocked by Hydroxy Urea treatment. From Sherwood and Sternberg, Dev Cell 2003, "To determine if the AC invasion signal generated by the 1 vulval lineage is linked to cell division, cell division was blocked with HU, and animals were analyzed just prior to and after the normal time for AC invasion. While no invasion was observed in animals at the early-to-mid L3 (20/20 animals), at the mid-to-late L3, 95% of ACs attached to the undivided 1fated P6.p cells (19/20 animals; Figure 5E). " This original result should be cited and stated. It would be fine to test cell division with another method further in this paper, but it is important to cite the original finding.

4. How many animals were examined for the results shown in Figure 3A and 3B (cyd::GFP and RNR::GFP)? Can this be stated in the figure legend or text?

5. I'm a little confused by Figure 4B. How many AC's were examined to make this graph? Can this be stated in the figure legend or graph?

6. The authors state, "It should be noted that the gap the AC created in the BMs of psf-1(lf) mutants could not be expanded because the underlying VPCs had ceased to proliferate (Fig. 1A bottom row)" However, in Figure 1D when the authors block P6.p cell division by expressing cki-1 in the P6.p, the BM breach looks wide. Can the breach in these two conditions be quantified? Perhaps the psf-1 does have an AC invasion defect?

7. The work in melanoma cell lines examining the replication-independent functions of MCM7,

by analyzing gene expression in cells arrested at the G1/S boundary after knockdown of MCM7 is interesting. While a role for MCM7 in promoting the expression of the PI3K came from this and was confirmed in C. elegans, where reciprocal findings also found? Specifically, did the work in melanoma cells also identify a role for MCM7 in promoting MMP expression as it does in the AC? It looks like this work might have found the opposite—" GO enrichment analysis of the molecular functions and biological processes of these core MCM7 target genes indicated that many of the up-regulated genes encode extracellular matrix, integrin, actin and ß-catenin interacting proteins". Can you comment/compare on the expression of MMPs in cancer cell lines and your analysis. This is helpful for the field, as pointing out differences as well as similarities is important in building a complete picture of this important and complex problem—pre-RC and cell invasion in different contexts.

8. Does PI3K signaling loss on its own result in an AC invasion defect? It is important that the authors perform AC-specific RNAi on age-1 and aap-1 to establish the requirements of PI3K signaling in the AC on its own. Many aspects of cell biological processes are redundant or provide robustness. This will provide broader insight into the role of the pre-RC into cell invasion, even if age-1 and aap-1 loss on their own don't result in an invasion defect.

9. Discussion section: "The PI3K pathway may be activated in the AC by the INA-2/PAT-2 ab-integrin complex on the invasive membrane [6]" It should be INA-1/PAT-3. There is no INA-2/PAT-2 integrin.

Rev. 2:

This submission reports a novel role for pre-replication complex (pre-RC) proteins (Orc2 and 5, MCM7, cdc6, cdt1) in the regulation of anchor cell invasion across the basement membrane during C. elegans development. The paper provides strong evidenced that this function is distinct from the known activity of pre-RC in the early stages of genome duplication, where they associate with chromatin to provide a scaffold for the initiation of DNA synthesis. The paper reports that in the C. elegans non-proliferating cells, pre-RC proteins act as transcriptional modulators, regulating the transcription of extracellular matrix genes to activating cell invasion upstream of the PI3 kinase pathway. This novel function of pre-RC components validates previous observations proposing that replication licensing proteins have other roles unrelated to cellular preparation for DNA synthesis.

The experimental approach is based on a screen for worm orthologs of genes overexpressed in melanoma exposed to migration stimulants. The experimental analyses utilize appropriate, state-of-the-art techniques. The results strongly support the hypothesis that Orc2 and 5, MCM7, cdc6, cdt1 play a role in the transcriptional regulation of genes involved in anchor cell invasion. The paper in its current form, however, does not address the question of whether these components perform this regulatory role as a part of a pre-RC complex similar to the one used in proliferating cells, or as components of other cellular multi-protein complexes. The paper in its current form also does not resolve if the helicase activity of pre-RC, or just the ATPase activity of MCM-7, is required for AC invasion.

1. The title and abstract of the current submission imply that the pre-RC, as a complex, participates in the transcriptional regulation of AC invasion. However, some pre-RC components that are essential for replication licensing do not seem to play a role in invasion, and the paper in its present form does not provide evidence that the genes identified in the screen act within the pre-RC complex. The reported observations could imply that some members of pre-RC play independent roles as parts of other molecular machines, a concept that already has a precedent (see #2 for references) as several members of the pre-RC complexes were reported to participate in cellular multi-protein complexes other than pre-RC, mediating a variety of cellular functions. This point should at least be discussed.

2. Precedents for interactions of pre-RC components with other multi-protein complexes that play a role in regulating cellular functions have been established (for example, for Drosophila ORC6 PMID: 12878722; centromeric association of ORC2, PMID: 15215892; for CDT1 at kinetochores PMID: 22581055 and PMID: 30154187). Appropriate citations should be included.

3. The expression patterns of MCM7 analyzed in the paper are interesting and support the conclusions that this protein plays a role in invasion. It would be interesting to follow the expression of other pre-RC components in a similar manner.

4. MCM-7 was the only component of the MCM helicase required for AC invasion, and the conclusion that the helicase activity of pre-RC plays a role in the regulation of AC invasion is based on the phenotype of MCM-7 variants harboring ATPase mutations. Although the ATPase activity of MCM-7 is essential for pre-RC helicase activity, the results observed in the current study do not rule out an independent and distinct role for MCM7 ATPase activity that is unrelated to its role in facilitating the MCM2-7 helicase. Because AAA ATPases often manifest more than a single function in cellular activity, this possibility should be discussed.

---

## [Decision Letter · Decision Letter 2]

18 Nov 2021

Dear Dr Hajnal,

Thank you for submitting your revised Research Article entitled "A DNA Replication-Independent Function of pre-Replication Complex Genes during Cell Invasion in C. elegans" for publication in PLOS Biology. I have now obtained advice from the original reviewers and have discussed their comments with the Academic Editor. 

The reviewers note that the additional data provided in the revision has further strengthened the manuscript and they appreciate the time and effort that was spent in addressing their concerns. Based on these reviews, we will probably accept this manuscript for publication, provided you satisfactorily address our remaining data and policy-related requests that I have provided below:

(A) You may be aware of the PLOS Data Policy, which requires that all data be made available without restriction: http://journals.plos.org/plosbiology/s/data-availability. For more information, please also see this editorial: http://dx.doi.org/10.1371/journal.pbio.1001797

- Supplementary files (e.g., excel). Please ensure that all data files are uploaded as 'Supporting Information' and are invariably referred to (in the manuscript, figure legends, and the Description field when uploading your files) using the following format verbatim: S1 Data, S2 Data, etc. Multiple panels of a single or even several figures can be included as multiple sheets in one excel file that is saved using exactly the following convention: S1_Data.xlsx (using an underscore).

- Deposition in a publicly available repository. Please also provide the accession code or a reviewer link so that we may view your data before publication.

Regardless of the method selected, please ensure that you provide the individual numerical values that underlie the summary data displayed in the following Figures, as they are essential for readers to assess your analysis and to reproduce it:

Figure 2D, 3D, 4B, 4D, 4F, 4H, 4J, 4L, 5A-B, 5D, 5F-H, S3B, S3D, S3F, S4C, S5B, S6B, S7E, S9A-C

(B) We note that Supplementary Figure S7A-B reports flow cytometry data. We ask that you please provide the raw FCS files of these data and recommend depositing the files in the FlowRepository database due to the potential size of the files (https://flowrepository.org/). If you do use the FlowRepository, please ensure that the files are publicly available at this stage and please provide the accession number/URL for the deposition. 

(C) Please deposit the underlying RNA-sequencing data (Figure S8) in a public database, such as the GEO. As before, please ensure that the data is made publicly available at this stage and that you provide the accession number/URL for the deposition. 

(D) Please also ensure that each of the relevant figure legends in your manuscript include information on *WHERE THE UNDERLYING DATA CAN BE FOUND*, and ensure your supplemental data file/s has a legend

(E) Please ensure that your Data Statement in the submission system accurately describes where your data can be found and is in final format, as it will be published as written there. This includes referencing where the underlying data can be found in the Supplementary Information, as well as providing the accession numbers for the data deposited in public databases.

(F) We require the original, uncropped and minimally adjusted images supporting all blot and gel results reported in an article's figures or Supporting Information files (Fig S7C-D). We will require these files before a manuscript can be accepted so please prepare and upload them now. Please carefully read our guidelines for how to prepare and upload this data: https://journals.plos.org/plosbiology/s/figures#loc-blot-and-gel-reporting-requirements

-------------------

We expect to receive your revised manuscript within two weeks.

*Published Peer Review History*

*Early Version*

Sincerely,

Richard

Richard Hodge, PhD

Associate Editor, PLOS Biology

rhodge@plos.org

Reviewer remarks:

Reviewer #1: I am very impressed with the thoughtful and rigorous response to my suggestions to clarify and improve the manuscript. The authors of this study have done an outstanding job with this work. Congratulations on the beautiful story-

Reviewer #2: The revision has addressed my concerns. I have no further comments.

---

## [Editor Report · Decision Letter 3]

1 Dec 2021

Dear Dr Hajnal,

On behalf of my colleagues and the Academic Editor, Hilary Coller, I am pleased to say that we can in principle accept your Research Article "A DNA Replication-Independent Function of pre-Replication Complex Genes during Cell Invasion in C. elegans" for publication in PLOS Biology, provided you address any remaining formatting and reporting issues. These will be detailed in an email that will follow this letter and that you will usually receive within 2-3 business days, during which time no action is required from you. Please note that we will not be able to formally accept your manuscript and schedule it for publication until you have any requested changes.

PRESS

Sincerely, 

Richard

Richard Hodge, PhD

Associate Editor, PLOS Biology

rhodge@plos.org

PLOS
